# Uni-MuMER: Unified Multi-Task Fine-Tuning of Vision-Language Model for Handwritten Mathematical Expression Recognition

**Yu Li**[*]    **Jin Jiang**[*]    **Jianhua Zhu**    **Shuai Peng**    **Baole Wei**    **Yuxuan Zhou**    **Liangcai Gao**[⊠]

Wangxuan Institute of Computer Technology, Peking University, Beijing, China

liyu@stu.pku.edu.cn    jiangjin@stu.pku.edu.cn    zhujianhuapku@pku.edu.cn    glc@pku.edu.cn

## Abstract

Handwritten Mathematical Expression Recognition (HMER) remains a persistent challenge in Optical Character Recognition (OCR) due to the inherent freedom of symbol layouts and variability in handwriting styles. Prior methods have faced performance bottlenecks by proposing isolated architectural modifications, making them difficult to integrate coherently into a unified framework. Meanwhile, recent advances in pretrained vision-language models (VLMs) have demonstrated strong cross-task generalization, offering a promising foundation for developing unified solutions. In this paper, we introduce Uni-MuMER, which fully fine-tunes a VLM for the HMER task without modifying its architecture, effectively injecting domain-specific knowledge into a generalist framework. Our method integrates three data-driven tasks: Tree-Aware Chain-of-Thought (Tree-CoT) for structured spatial reasoning, Error-Driven Learning (EDL) for reducing confusion among visually similar characters, and Symbol Counting (SC) for improving recognition consistency in long expressions. Experiments on the CROHME and HME100K datasets show that Uni-MuMER achieves super state-of-the-art performance, outperforming the best lightweight specialized model SSAN by 16.31% and the top-performing VLM Gemini2.5-flash by 24.42% under zero-shot setting. Our datasets, models, and code are open-sourced at: https://github.com/BFlameSwift/Uni-MuMER

## 1 Introduction

Handwritten Mathematical Expression Recognition (HMER) seeks to translate handwritten expressions into machine-readable markup, supporting document understanding and digital preservation of scientific material. Unlike standard Optical Character Recognition (OCR), HMER involves parsing complex 2D structures [15], ambiguous symbols, and the inherent freedom of symbol layout and variability in handwriting styles [68], requiring not just recognizing individual symbols but also layout reasoning and parsing their complex spatial relationships. Recent approaches have primarily leveraged RNNs [23] or Transformers [48] due

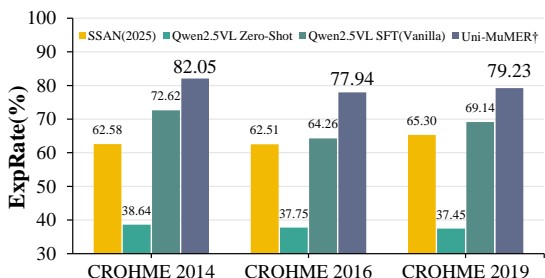

Figure 1: **Performance comparison on CROHME sets (%).** Our Uni-MuMER[†] achieves significant improvements compared to the previous SOTA (SSAN) and Qwen2.5VL Zero-Shot.

---

[*] Equal contribution,   [⊠]Corresponding authors.

39th Conference on Neural Information Processing Systems (NeurIPS 2025).

to their powerful sequential modeling capabilities. Additional modules, such as tree-structured decoders [64, 52, 70] and relative-position-aware mechanisms [18, 62, 53, 54], have been introduced to further enhance performance. These dominant research directions in HMER have focused on incorporating prior human knowledge into model architectures, particularly through manually designed structural modules or attention mechanisms [63, 65, 66, 67, 30, 18, 4].

Despite the importance of HMER and numerous proposed improvements, the field has seen only marginal progress in recent years. As is shown in Tab. 1, performance on the CROHME datasets [38, 39, 37] has improved by merely 3% (from CoMER [66] to SSAN [62]), underscoring an urgent need for a novel paradigm. The limited progress of these models arose from three constraints: (1) Improvements are isolated and model-specific, making them hard to integrate or scale. (2) Optimizing across multiple auxiliary tasks remains challenging, with many approaches focusing on singular priors rather than adopting a unified, multi-faceted enhancement [7]; and (3) Models trained on single-domain datasets, struggling to generalize to other datasets, lack essential scalability and transferability [68, 46]. Additionally, widely standard metrics like ExpRate fail to capture visual equivalence in LaTeX outputs, and the scarcity of diverse data exacerbates these constraints. Recently introduced visual metrics such as CDM and large-scale datasets like Mathwriting offer opportunities for large-scale cross-dataset training and multiple LaTeX syntax style evaluation.

Rapid advancements in pretrained vision-language models (VLMs) have significantly enhanced foundational recognition capabilities and generalization across related tasks [8, 25, 28, 31, 32]. While early benchmarks like OCRBench [34] reported poor HMER performance for generalist VLMs, Tab. 1 presents our evaluation of recent open-source and closed-source VLMs, such as Qwen2.5-VL [3], Doubao-1.5-pro [45], Gemini2.5-flash [11], and GPT4o [41]. Our results indicate that these large-scale models have exhibited unexpectedly strong capabilities in handling structured recognition tasks. However, these high-performance closed-source models trained on massive amounts of undisclosed data [1] make it difficult to pinpoint how to systematically improve performance on HMER. Consequently, empowering open-source VLMs to achieve comparable or superior HMER performance remains an open and urgent challenge.

To bridge this gap, we propose Uni-MuMER, a unified multi-task fine-tuning framework for enhancing open-source VLMs in HMER. Unlike previous methods constrained by single-domain datasets or isolated architectural improvements, Uni-MuMER fully exploits existing data resources through data-driven fine-tuning. Another motivation of our work is to unify previously fragmented HMER tasks into a unified framework, shifting the focus from incremental architectural modifications toward generalizable recognition capabilities. Concretely, we introduce specialized training data and learning objectives, employing novel tasks such as Tree-Aware Chain-of-Thought for explicit structural reasoning [51, 57], Error-driven learning to reduce symbol confusion [36], and Symbol Counting [30] to enhance parsing expressions capabilities. Fig. 1 nicely illustrates the jump with Uni-MuMER; it exceeds Qwen-2.5-VL in the zero-shot setting and the vanilla SFT setting across multiple CROHME datasets. Besides, owing to advancements in open-source inference frameworks (e.g., vllm [29]), our method achieves superior inference speeds compared to traditional methods, enhancing its practical applicability. Our contributions are listed below:

- We propose a new unified paradigm for HMER. Unlike prior methods that heavily rely on specialized networks and single-task training, our data-driven multi-task method injects domain knowledge into a generalist VLM, yielding cumulative performance gains.
- We introduce three data-driven tasks: Tree-Chain-of-Thought, Error-Driven Learning, and Symbol Counting. These systematically address the challenges of HMER, which are two-dimensional structure, ambiguous handwriting, and output consistency.
- Our Uni-MuMER method achieves new SOTA results on the CROHME and HME100K datasets. Notably, it attains 79.74% averaged across CROHME datasets (+41.79% over Qwen2.5-VL, +24.42% over Gemini2.5-flash in zero-shot setting, and +16.31% over specialized models SSAN).

## 2 Related Work

### 2.1 HMER

Early rule-based approaches attempted to address the challenges of HMER through symbol segmentation, recognition, and syntactic parsing based on handcrafted grammars [6, 60, 10, 2, 22, 47]. These methods struggled with generalization to diverse handwriting styles and complex layout structures.

**Sequence-based decoding methods** The emergence of deep learning shifted HMER towards end-to-end sequence modeling tasks. Early encoder-decoder architectures based on RNNs, WAP [65] first applied sequence-to-sequence learning to HMER. Subsequent RNN-based models improved visual encoding [63] and robustness to handwriting style [53, 54]. Inspired by the Transformer's success [48], BTTR [67] introduced the first Transformer-based model for HMER. To enhance alignment during decoding, CoMER [66] introduced a coverage attention mechanism.

**Multi-task Learning** Beyond sequence modeling, researchers have explored structural decoders and multi-task models to better capture 2D structure. One central line of work focuses on modeling the hierarchical tree structure of expressions. TreeDecoder (TD) [64] and its improved version TDv2 directly predict a tree-structured representation of the expression. SAN [58] introduced syntactic constraints into the decoding process, and TAMER [70] jointly optimizes both sequence and tree decoding within a unified Transformer framework. Various auxiliary tasks have also been proposed to inject domain knowledge: ABM [4] adds a dual-direction decoder loss to enhance context modeling. RLFN [9] fuses a language-model-based module with the recognizer to leverage LaTeX syntax and context. ICAL [69] proposed an implicit character construction to capture latent symbol-level semantics. To address symbol omission and repetition, CAN [30] incorporated an auxiliary symbol-counting task. UniMERNet [49] adds a Length-Aware Module targeting real-world expressions' extreme length variance. PosFormer [18] and SSAN [62] explicitly model spatial relationships among symbols to guide the network's attention.

## 2.2 Vision-Language Models

Vision-Language Models (VLMs), initially popularized by contrastive learning frameworks like CLIP [43], laid the groundwork for powerful zero-shot multimodal recognition. Donut [27] and LayoutLMv3 [24] extended this capability specifically to OCR tasks, effectively recognizing diverse textual content. High-resolution, document-centric VLMs, notably Monkey [33] and TextMonkey [35], enhanced OCR performance by combining patch-based image encoding with explicit textual supervision. For mathematical OCR, models like Im2LaTeX [12] and Nougat [5] focused explicitly on end-to-end LaTeX reconstruction from printed equations and scientific documents. The recent FERMAT benchmark [40] explicitly evaluated multiple SOTA VLMs on handwritten mathematics recognition, revealing critical gaps between general text OCR and handwritten mathematical recognition. Domain-specific adaptations such as VLPG [20] and HiE-VL [21] propose graph-based or hierarchical adapters to enhance mathematical recognition, although their extensive architectural modifications and modest performance limit broader adoption. In contrast, recent general-purpose frameworks like Qwen2.5-VL [3], equipped with dynamic-resolution ViTs and structured outputs, demonstrate promising potential for adaptation to mathematical OCR tasks.

## 3 Method

We introduce Uni-MuMER, a unified multi-task fine-tuning framework for enhancing open-source VLMs in HMER. The overall pipeline is shown in Fig. 2. Given an input image of a handwritten expression and a task-specific instruction, the model is trained to produce the corresponding output. Uni-MuMER integrates four tasks: Vanilla HMER, Tree-aware Chain-of-Thought, Error-Driven Learning, and Symbol Counting, into a unified training paradigm. The primary goal is to adapt a general-purpose VLM to the highly structured and domain-specific knowledge of HMER without modifying any original architecture. Subsequent subsections provide additional details for each task.

### 3.1 Vanilla HMER

**Base Model** We adopt Qwen2.5-VL-3B [3] as the VLM Backbone. It comprises a ViT-based visual encoder and a transformer-based language decoder, pre-trained to perform various image-to-text tasks, which provide robust visual grounding and structured sequence-generation capabilities, rendering it an effective foundation for end-to-end fine-tuning in HMER tasks.

**Vanilla HMER** As is shown at the top of Fig. 2, Vanilla HMER involves providing an image of a mathematical expression alongside a textual instruction, prompting the model to directly output the corresponding LaTeX formatted expression. Traditional lightweight specialized models took images alone as input and generated the expressions, whereas VLMs now exhibit strong generalization capabilities across related tasks. We also utilize models fine-tuned specifically on the Vanilla HMER

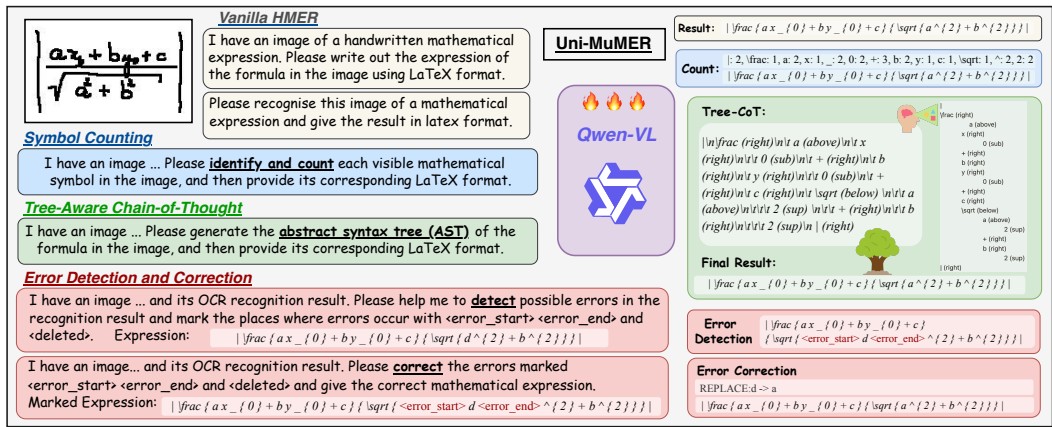

Figure 2: **Overview of our Uni-MuMER Framework**: which augments a VLM base with four distinct integrated tasks: Vanilla HMER, Tree-Aware Chain-of-Thought, Error-Driven Learning, and Symbol Counting to generate robust and accurate LATEX from an input image of a handwritten expression.

task as baselines for subsequent comparison. Through Vanilla HMER, the model acquires essential recognition capabilities and ensures accurate and structured outputs.

## 3.2 Tree-Aware Chain-of-Thought

We propose a Tree-CoT task that employs parsing Abstract Syntax Trees (ASTs) as intermediate cognitive structures for expression recognition. The model accepts an input comprising an image and an instruction, subsequently generating both a serialized tree and the final recognized LATEX expression. Inspired by ASTs widely utilized in mathematical information retrieval [61, 42] and previous tree-decoding methods in HMER [64, 52, 70], we incorporate explicit tree-structured decoding into the textual output. This explicit intermediate representation enables the model to directly reason the inherent 2D spatial relationships among symbols. Consequently, Tree-CoT guides the model to produce structurally coherent and semantically accurate LATEX outputs by bridging visual input processing and linear textual serialization. Below, we describe in detail the construction.

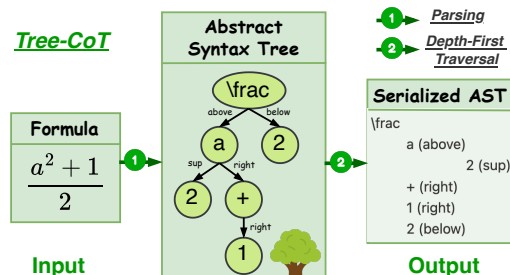

Figure 3: **Illustration of the proposed Tree-CoT construction and serialization procedure.** For input expression, we parse it into an AST, then traverse in depth-first order and serialize it into a structured textual format.

**Tree-CoT** The construction procedure of Tree-CoT is illustrated in Fig. 3. We first parse the input expression to derive its Abstract Syntax Tree (AST), explicitly capturing hierarchical and spatial relationships among symbols. We subsequently perform a depth-first traversal (DFS) of the AST to sequentially linearize the tree structure. To encode this structured representation, we introduce a specific serialization scheme, using tab-based indentation to reflect tree depth and newline separation to distinguish individual nodes clearly, and the raw text is in Fig. 2. Each serialized line explicitly contains the symbol label and its spatial relation to its immediate parent node, resulting in a coherent textual representation of the AST.

## 3.3 Error-Driven Learning

Error-Driven Learning (EDL) leverages a learning-from-mistakes paradigm to enhance model accuracy. Its input consists of an error corpus generated by the model itself, structured into two distinct subtasks: error detection and error correction. Inspired by self-improvement strategies and previous

work indicating the suitability of language models for correcting HMER errors [9], we integrate these subtasks explicitly. The subsequent sections detail the generation of the error corpus and the definitions of error detection and correction subtasks.

**Error Corpus Generation** As is shown in the Fig. 4. Initially, the complete dataset (e.g., CROHME) is randomly partitioned into multiple distinct subsets denoted as $F_1, F_2, \ldots, F_n$. Subsequently, we perform cross-validation training, where each VLM is trained on $n-1$ folds and evaluated on the remaining held-out fold $F_i$. During this, multiple candidate predictions are generated through multiple sampling for each input image to collect potential model outputs. By comparing these candidate predictions to their corresponding ground-truth labels, we explicitly identify erroneous outputs and compile these erroneous predictions into the final error corpus.

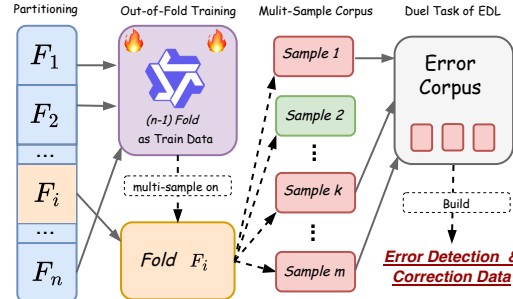

Figure 4: **Illustration of Error Corpus Generation for Error-Driven Learning Task (EDL)**, where erroneous model outputs are collected through cross-fold training and multiple sampling from held-out evaluation subsets.

**Error Detection and Correction** Based on this error corpus, we formulate two related subtasks: error detection and error correction. As shown in the Fig. 2, the error-detection task takes as input the original expression image alongside a potentially erroneous predicted expression, outputting a marked expression wherein errors are explicitly enclosed by `<error_start>` and `<error_end>` tags, and omissions are explicitly indicated by `<deleted>` tags. The subsequent error-correction subtask receives this masked expression as input and produces the correction log and corresponding LaTeX expression. The scale of the error corpus obtained for each training set used in Sec. 4.1 is approximately equal to that of the original training data.

### 3.4 Symbol Counting Auxiliary Task

We introduce a Symbol Counting (SC) auxiliary task designed to encourage the model to explicitly account for all symbols appearing in expressions. The input to this task comprises an expression image along with a counting instruction, and the expected output consists of the total count of visible symbols in the expression alongside the corresponding LaTeX expression. Inspired by the CAN [30], we observe that models frequently produce locally coherent yet globally inconsistent outputs, especially involving repeated symbols. To mitigate this issue, we explicitly integrate symbol-counting information into the textual output representation. This task compels the model to accurately predict symbol counts, thus reducing the likelihood of symbol omissions or hallucinations during final LaTeX expression generation.

**Symbol Counting.** We construct training targets that prepend the count string to the actual LaTeX. Specifically, for a given handwritten expression image, the target output becomes `<Count String>\n<Expression string>`. For example: For the expression $\frac{a^2+1}{2}$, the textual count string is: `\frac:1,a:1,2:2,+:1`.

## 4 Experiments

### 4.1 Experimental Setup

This section details the experimental setup used to evaluate the Uni-MuMER, including datasets, evaluation metrics, training settings, and baselines.

**Datasets** The **CROHME** series [37, 38, 39] is the most widely used dataset for HMER. It comprises 8836 training samples, and three test sets contain 986, 1147, and 1199 images for the CROHME 2014, 2016, 2019. Recently, the **CROHME 2023** [56] iteration provided an expanded training set of 10979 images and a test set of 2300 images. **HME100K** [59] is a large-scale, real-world dataset. It consists of 74502 training and 24607 testing images, encompassing various writing styles and conditions. **MathWriting** [17], released by Google Research in 2024, is the largest HME corpus to date with 230K human-written and 400K synthetically-generated expressions. Finally, **Im2Latexv2** [44] builds upon the printed expression dataset Im2Latex-100k[12] and enhances it with improved LaTeX normalization and 59 diverse rendering styles for more realistic expression images.

Table 1: **Performance comparison on CROHME datasets(%).** We evaluate our model and four baselines using expression recognition rate (Exp) and ExpRate@CDM (@CDM) on the CROHME 2014, 2016 and 2019 test sets. We use the following notation for reported results: All models are evaluated under zero-shot inference settings, [a] denotes training with data augmentation, [p] denotes reproduced by us, and [†] denotes the use of external data.

| Method | CROHME14 | | CROHME16 | | CROHME19 | | Average | |
|---|---|---|---|---|---|---|---|---|
| | Exp↑ | @CDM↑ | Exp↑ | @CDM↑ | Exp↑ | @CDM↑ | Exp↑ | @CDM↑ |
| *Closed-Source Models* | | | | | | | | |
| GPT-4o[41] | 50.61 | 60.85 | 46.03 | 50.65 | 49.79 | 57.13 | 48.81 | 56.21 |
| Doubao-1.5-pro[45] | 50.41 | 69.17 | 49.00 | 64.08 | 46.87 | 64.05 | 48.79 | 65.77 |
| Gemini2.5-flash [41] | 58.01 | 69.98 | 52.74 | 61.34 | 55.21 | 65.38 | 55.32 | 65.57 |
| *Open-Source Models* | | | | | | | | |
| Qwen2.5-VL-3B[3] | 38.64 | 44.01 | 37.75 | 43.33 | 37.45 | 43.04 | 37.95 | 43.46 |
| Qwen2.5-VL-7B[3] | 55.98 | 65.51 | 50.92 | 56.32 | 49.62 | 56.55 | 52.17 | 59.46 |
| Qwen2.5-VL-72B[3] | 59.74 | 68.56 | 54.32 | 59.37 | 55.13 | 61.05 | 56.40 | 62.99 |
| *Specialized Models* | | | | | | | | |
| DenseWAP[63] | 50.1 | – | 47.5 | – | – | – | – | – |
| BTTR[67] | 53.96 | – | 52.31 | – | 52.96 | – | 53.08 | – |
| ABM[4] | 56.85 | – | 52.92 | – | 53.96 | – | 54.58 | – |
| CAN-ABM[30] | 57.26 | – | 56.15 | – | 55.96 | – | 56.46 | – |
| CoMER[66] | 58.38 | 61.66 | 56.98 | 63.03 | 59.12 | 62.71 | 58.16 | 62.47 |
| ICAL | 60.63 | – | 58.79 | – | 60.51 | – | 59.98 | - |
| PosFormer[18] | 60.45 | – | 60.94 | – | 62.22 | – | 61.20 | – |
| TAMER[70] | 61.36 | 63.28 | 59.54 | 62.94 | 60.13 | 63.46 | 60.34 | 63.23 |
| SSAN[62] | 60.85 | – | 60.02 | – | 61.83 | – | 60.90 | – |
| CoMER[a][66] | 59.33 | – | 59.81 | – | 62.97 | – | 60.70 | – |
| CoMER[ap†][66] | 46.96 | 53.55 | 43.33 | 48.47 | 48.25 | 54.84 | 46.18 | 52.29 |
| PosFormer[a][18] | 62.74 | 65.88 | 61.03 | 65.39 | 64.97 | 69.30 | 62.91 | 66.85 |
| SSAN[a][62] | 62.58 | – | 62.51 | – | 65.30 | – | 63.43 | – |
| UniMERNet[a†] | 67.4 | – | 68.4 | – | 65.4 | – | 67.07 | – |
| VLPG[†][20] | 60.41 | – | 60.51 | – | 62.34 | – | 61.09 | – |
| HiE-VL[a†][21] | 73.30 | – | 70.70 | – | 69.3 | – | 71.13 | – |
| **Vanilla (baseline)** | 72.62 | 75.66 | 64.26 | 67.83 | 69.14 | 71.98 | 68.64 | 71.82 |
| **Uni-MuMER** | 75.36$^{+2.74}$ | 79.11$^{+3.45}$ | 70.79$^{+6.53}$ | 75.24$^{+7.41}$ | 73.73$^{+4.59}$ | 77.23$^{+5.25}$ | 73.29$^{+4.65}$ | 77.19$^{+5.37}$ |
| **Uni-MuMER[†]** | **82.05**$^{+9.43}$ | **85.09**$^{+9.43}$ | **77.94**$^{+13.68}$ | **81.08**$^{+13.25}$ | **79.23**$^{+10.09}$ | **82.40**$^{+10.42}$ | **79.74**$^{+11.10}$ | **82.86**$^{+11.04}$ |

**Evaluation Metrics** The standard metric for HMER is **Expression Recognition Rate (ExpRate)**, which measures exact-match accuracy. However, Exprate unfairly penalizes visually correct predictions that differ only in LaTeX syntax styles. **Character Detection Matching (CDM)** offers a visual evaluation alternative. It renders both prediction and ground-truth into images, comparing them via character detection that considers spatial layout. This yields two metrics: the CDM Score, quantifying visual similarity, and Exprate@CDM, a visually-based accuracy measure. These CDM metrics provide evaluation robustness against variations in LaTeX syntax.

**Training** We fine-tune the Qwen2.5-VL-3B model on all datasets with three data-driven tasks simultaneously for a single epoch. Further details can be found in the Appendix B.4.

**Baselines** We compare our methods against four types of baselines: (i) Closed-source VLMs, including Gemini2.5-flash, etc; (ii) Open-source VLMs, such as the Qwen-2.5VL family. Both closed-source VLMs and open-source VLMs are evaluated under zero-shot inference settings; (iii) Previous SOTA methods on HMER and their data augmentation versions, for example, PosFormer and SSAN, etc; and (iv) A Vanilla(baseline) and a Uni-MuMER model, which is only fine-tuned on the CROHME training set to ensure a fair comparison with previous SOTA methods.

## 4.2  Main Results

Tab. 1 displays the strong performance of our Uni-MuMER and previous methods on the CROHME dataset. As CDM is a recently proposed metric, we reproduced all reported results involving CDM. For a fair comparison, the Vanilla (baseline) reported in Tab. 1 and Tab. 2 was trained exclusively on either CROHME or HME100K dataset. The Uni-MuMER variant incorporated training samples constructed from the corresponding images for three tasks. Further utilizing external data, UniMuMER[†] was trained on approximately 1.6M samples, comprising data from 386K images across the three tasks and all datasets. Due to space limitations, additional results are presented in Appendix D.

**Comparison with Previous SOTA Methods** Data augmentation has traditionally improved performance in lightweight models, yet inherent limitations still restrict overall effectiveness. Recently, VLPG [20] and HiE-VL [21], have integrated VLM into HMER. HiE-VL enhances performance through non-end-to-end training, architectural refinements, and extra data. However, both Uni-MuMER and Uni-MuMER[†] (use external data) maintain a notable performance advantage over these VLM-based methods. Notably, we also trained CoMER[a†] [66] using the same external data (386K images) and observed a significant performance degradation. This finding indicates that the inherent limitations of lightweight models hinder their ability to effectively utilize large and diverse datasets.

**Comparison with Large VLMs (Zero-Shot)** Next, we investigate the zero-shot performance of both open-source (Qwen-2.5VL family) and closed-source large VLMs on CROHME. Although open-source models exhibit nontrivial results, they tend to lag behind previous SOTA methods when no fine-tuning is involved. Contrastingly, certain closed-source VLMs like Gemini2.5-flash exhibit remarkable accuracy, surpassing lightweight specialized models in a purely zero-shot setting. They still underperform relative to our proposed Uni-MuMER and Uni-MuMER[†] method.

**Incremental Gains** Building upon the Qwen2.5-VL-3B VLM base, our Vanilla (baseline) model already surpasses most published SOTA methods. After integrating three data-driven tasks, Tree-CoT, EDL, and SC, our method, Uni-MuMER[†] further improves both ExpRate and ExpRate@CDM, as shown in Tab. 1. For a fair comparison, both the Vanilla (baseline) and Uni-MuMER are trained solely on the CROHME training set, without using any external data. Introducing Upon incorporating external data along with these tasks (1.6M samples), our final model, Uni-MuMER[†], achieves even greater performance, notably reaching an ExpRate of 79.74% and ExpRate@CDM of 82.86% on the CROHME Average. Both Uni-MuMER and Uni-MuMER[†] significantly outperform both the Vanilla (baseline) model and prior methods, underscoring the vital role of data-driven tasks and diverse external data in enhancing model accuracy.

**Performance on HME100K** To focus the evaluation on mathematical expression recognition, following [16], we exclude all test instances containing CJK characters. Detailed impacts are discussed in Appendix C.5. Tab. 2 shows that open-source and closed-source VLMs exhibit modest performance, which is due to the challenging real-world conditions in HME100K dataset. In contrast, several prior lightweight specialized methods exclusively trained on HME100K, such as TAMER, achieve strong performance. Although HiE-VL utilizes external training data, it achieves limited results on HME100K, even underperforming compared to prior models. Our Uni-MuMER achieves SOTA performance. Additionally, by using external data, Uni-MuMER[†] further enhances its performance, clearly surpassing HiE-VL.

Table 2: **Performance on HME100K dataset (%),** evaluated by Exp, @CDM, and CDM. The [†] denotes the use of external data.

| Method | HME100K | |
|---|---|---|
| | Exp↑ | @CDM↑ |
| GPT-4o [41] | 22.96 | 27.02 |
| Gemini2.5-flash [11] | 28.14 | 34.32 |
| Doubao1.5-pro [45] | 43.90 | 55.08 |
| Qwen2.5-VL-3B [3] | 44.42 | 47.41 |
| Qwen2.5-VL-7B [3] | 54.57 | 58.84 |
| Qwen2.5-VL-72B [3] | 49.59 | 55.09 |
| DenseWAP [63] | 61.85 | - |
| CoMER [66] | 68.12 | 70.36 |
| TAMER [70] | 69.50 | 71.30 |
| HiE-VL[†] [21] | 64.20 | - |
| Vanilla (baseline) | 70.15 | 71.80 |
| Uni-MuMER | 71.62[+1.47] | 73.40[+1.60] |
| Uni-MuMER[†] | **72.66**[+2.51] | **74.30**[+2.50] |

## 5 Analysis

### 5.1 Ablation Study

In this section, we present an ablation study to quantify the contribution of the three proposed tasks, Tree-CoT, EDL, and SC. As is shown in Tab. 3, each individual task enhances performance compared to the Vanilla(baseline). The Uni-MuMER combined integration of all tasks results in the best overall performance, while removing any module leads to noticeable performance degradation, highlighting their complementary roles and collective importance.

Table 3: **CROHME-Average ablation results (ExpRate %).**

| Tree-CoT | EDL | SC | ExpRate↑ |
|---|---|---|---|
| ✗ | ✗ | ✗ | **68.64** |
| ✓ | ✗ | ✗ | **70.85**[+2.21] |
| ✗ | ✓ | ✗ | **70.30**[+1.66] |
| ✗ | ✗ | ✓ | **69.86**[+1.22] |
| ✗ | ✓ | ✓ | **71.95**[+3.31] |
| ✓ | ✗ | ✓ | **71.06**[+2.42] |
| ✓ | ✓ | ✗ | **71.76**[+3.12] |
| ✓ | ✓ | ✓ | **73.29**[+4.65] |

**Tree-CoT enhances comprehension of structurally complex expressions** Fig. 5a presents a comparative analysis of accuracy across models with varying structural complexities of expressions. The Vanilla model experiences a significant degradation in accuracy as structural complexity increases. Introducing the Tree-CoT strategy notably alleviates this performance decline. Specifically, for structurally complex expressions, Vanilla+Tree-CoT demonstrates an accuracy improvement of approximately 5–6% over the Vanilla baseline, closely matching the performance of the Full model. Conversely, Tree-CoT offers marginal improvements for simpler expressions, suggesting its primary utility is enhancing model robustness and accuracy when handling structurally intricate expressions.

**EDL reduces confusion among visually similar characters** Tab. 4 enumerates the top five most frequent alphabet-number confusion pairs on CROHME-Average for the Vanilla model and Vanilla+EDL. The Vanilla model frequently misclassifies pairs such as 2↔z, resulting in 5.24 misrecognitions per expression in all SUB1 cases. After applying EDL, the number of misclassifications decreases to 3.31. This implies that EDL effectively guides the model in distinguishing fine-grained differences among commonly confused symbols, enhancing symbol-level accuracy and overall recognition performance.

Table 4: **Top-5 alphabet–number confusions (SUB1) on CROHME-Average, comparing the Vanilla vs. Vanilla + EDL.**

| Type | Top-5 misrecognition pair (%) | | | | | $\Sigma$ |
| | $2 \leftrightarrow z$ | $0 \leftrightarrow o$ | $3 \leftrightarrow z$ | $1 \leftrightarrow i$ | $1 \leftrightarrow n$ | |
| --- | --- | --- | --- | --- | --- | --- |
| *Vanilla* | 1.40 | 1.48 | 0.74 | 0.89 | 0.74 | 5.25 |
| *w/ EDL* | 1.23 | 1.16 | 0.15 | 0.62 | 0.15 | 3.31 |
| $\Delta$ ($\downarrow$) | −0.17 | −0.32 | −0.58 | −0.27 | −0.59 | 1.94 |

**SC improves recognition consistency in long expressions** Fig. 5b compares model accuracy with increasing symbol repetition. The Vanilla baseline accuracy deteriorates as repetition frequency rises, when symbols repeat five or more times. Incorporating the SC task substantially mitigates this decline, closely matching the Uni-MuMER model's stable performance under high repetition conditions. However, for simpler expressions with minimal repetition, SC slightly reduces accuracy compared to the Vanilla baseline, suggesting it occasionally diverts attention from primary recognition tasks.

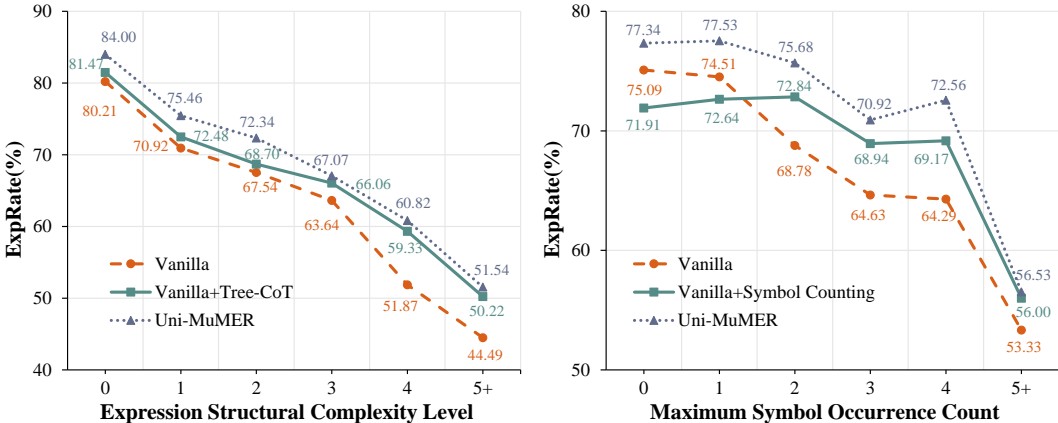

(a) Model performance relative to structural complexity. The Tree-CoT task notably reduces accuracy loss as structural complexity increases.

(b) Model performance concerning symbol repetition count. Symbol Counting (SC) task mitigates accuracy degradation caused by frequent symbol repetitions.

Figure 5: **Model accuracy comparison under varying (a) structural complexities and (b) symbol repetition counts.** Both models significantly outperform the Vanilla baseline, approaching Uni-MuMER model performance in challenging scenarios.

## 5.2 Effects of Data Diversity

In this section, we examine the influence of incrementally scaling training datasets on model performance. As illustrated in Fig. 6, start with the CROHME dataset, which serves as the standard benchmark in HMER. We then sequentially incorporate increasingly diverse datasets, including the updated CROHME-2023 set, the real-scene HME100K dataset, the extensive handwritten corpus MathWriting, and finally the printed-expression Im2LaTeXv2 dataset. At each step, the model is retrained from scratch on the cumulative training data and evaluated across multiple distinct test sets.

To effectively account for variations in LATEX notation styles, we employ ExpRate@CDM as the standardized metric for consistent performance comparison.

**Benefits of Enhanced Data Diversity** Each incremental expansion of the training sets with more varied handwritten expressions yields consistent gains on all evaluation sets, confirming that performance scales with data diversity.

**Out-of-Domain Generalization** Accuracy on printed expression images gradually improves with increased handwritten training data, reflecting partial transfer of structural knowledge for expression recognition. Nevertheless, its accuracy improves significantly after adding Im2LaTeXv2, reflecting the value of in-domain printed data for that target.

**Leave-One-Out Ablation** To further investigate the out-of-domain generalization performance, we conducted an additional "Leave-One-Out" experiment, training Uni-MuMER on N-1 datasets and evaluating it on the held-out N-th dataset.

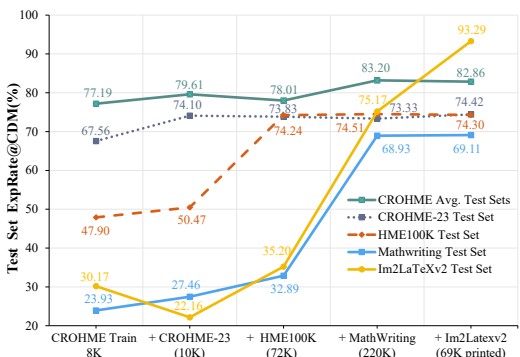

Figure 6: **Impact of incrementally training data scaling on ExpRate@CDM.** Starting from the CROHME training set, we incrementally incorporate four additional datasets. The model is retrained from scratch and evaluated on five distinct test sets at each expansion step.

The results below present the expression recognition accuracy in each scenario. For comparison, we include the baseline CoMER and Doubao-1.5-pro under zero-shot settings.

Table 5: **Leave-one-out ablation of training datasets. (ExpRate@CDM%)** We train Uni-MuMER in six configurations: using all data (all five sources), and using all-minus-one for each major dataset. Results are reported on CROHME-Average, CROHME 2023, HME100K, MathWriting, and Im2LaTeXv2. For reference, we also show a specialized model (CoMER) trained with and without CROHME, and a closed-source VLM (Doubao-1.5-pro) for reference.

| Model | CROHME-Avg | CROHME 2023 | HME100K | MathWriting | Im2LaTeXv2 |
|---|---|---|---|---|---|
| CoMER[†] (w/o CROHME) | 45.99 | 58.28 | 39.49 | 23.23 | 40.39 |
| CoMER[†] | 52.29 | 59.91 | 44.63 | 28.45 | 53.37 |
| Doubao-1.5-pro | 65.77 | 58.18 | 55.08 | 26.34 | 27.66 |
| Uni-MuMER[†] (w/o CROHME) | *78.22* | 72.19 | 74.47 | 68.08 | 89.18 |
| Uni-MuMER[†] (w/o CROHME-2023) | 82.31 | *71.66* | 74.68 | 68.85 | 89.77 |
| Uni-MuMER[†] (w/o HME100K) | 82.30 | 72.01 | *52.38* | 68.94 | 91.19 |
| Uni-MuMER[†] (w/o MathWriting) | 79.58 | 73.51 | 74.29 | *32.55* | **93.31** |
| Uni-MuMER[†] (w/o Im2LaTeXv2) | **83.20** | 73.33 | **74.51** | 68.93 | *75.17* |
| Uni-MuMER[†] | 82.89 | **74.42** | 74.30 | **69.11** | 93.29 |

The leave-one-out evaluation confirms that our unified model can generalize to an unseen dataset reasonably well. When the unseen domain is very different, performance drops more markedly, which highlights an area for improvement. HME100K for real-life, low-quality images, MathWriting for densely structured expression, and Im2LaTeXv2 for printed expression images, each of which represents a markedly different and challenging domain.

Crucially, after introducing even modest amounts of domain-relevant data, our unified approach easily achieves top-tier performance. Thus, our results indicate that, compared to existing specialized models and general VLM baselines, our approach provides superior generalization with minimal domain-specific data requirements, highlighting a clear advantage and direction for future improvement.

## 5.3 Mixing General-Purpose VLM Data

From the perspective of general-purpose LLMs and VLMs, mixing data from different domains can yield substantial performance gains. Although our Uni-MuMER addresses previous limitations by introducing a unified solution, achieving super SOTA performance, it's necessary to validate whether Uni-MuMER benefits from general domain data beyond HMER-only training and how Uni-MuMER-Data affects the model's general capabilities. To this end, we introduce Uni-MuMER-LLAVA, fine-tuned using an equal 1:1 mixture of HMER and LLaVA-OneVision data, and compare its performance with Uni-MuMER across benchmarks assessing HMER-specific and general capabilities.

Table 6: **Comparison among Uni-MuMER, Uni-MuMER-LLAVA and Qwen2.5-VL on multi-modal reasoning benchmarks.** Results are reported on MMMU_val, Math-Vision, and Math-Vista_testmini, covering general multimodal understanding and mathematical reasoning tasks.

| Model | MMMU_val | Math-Vision | MathVista_testmini |
|---|---|---|---|
| Uni-MuMER | 47.89 | 24.01 | 47.8 |
| Uni-MuMER-LLAVA | 48.67 | **24.34** | **51.1** |
| Qwen2.5-VL-3B-Instruct | **52.78** | 21.38 | 47.9 |

Table 7: **Cross-benchmark comparison between Uni-MuMER and Uni-MuMER-LLAVA across benchmarks.** Results are reported on CROHME-Avg, CROHME-2023, HME100K, MathWriting, and Im2LaTeXv2.

| Model | CROHME-Avg | CROHME-2023 | HME100K | MathWriting | Im2LaTeXv2 |
|---|---|---|---|---|---|
| Uni-MuMER | **82.89** | **74.42** | **74.30** | 69.11 | 93.29 |
| Uni-MuMER-LLAVA | 81.99 | 69.10 | 73.48 | **70.48** | **93.44** |

As shown in Tab. 6 and Tab. 7, Uni-MuMER exhibits strong generalization capabilities despite being fine-tuned exclusively on HMER-specific data, achieving performance comparable to Qwen2.5VL-3B. Notably, Uni-MuMER-LLAVA further enhances overall performance, surpassing Uni-MuMER on general tasks like MMMU and Math, while incurring marginal performance decreases on specific HMER test sets. It's promising to integrate our method and Uni-MuMER-Data to empower VLMs.

## 6 Discussion

**For Small Models: Larger Model, Better Performance, Faster Speed** Previous HMER models were lightweight and relied on task-specific architectural designs (e.g., tree-structured modules). Uni-MuMER builds upon both these task-specific insights and current VLM, taking a significant step forward with a unified and more powerful model. Benefiting from recent advances in efficient inference frameworks, Uni-MuMER not only achieves substantially improved performance but also delivers faster inference compared to smaller models. Further detail is provided in Appendix C.2.

**For VLM: A Chain-of-Thought Perspective on Expression Recognition** Unlike direct expression recognition, Uni-MuMER adopts a chain-of-thought perspective by formulating three tasks: expression tree construction, error detection and correction, and symbol counting. These tasks guide the model step by step toward the final prediction, enhancing both interpretability and generalization. This approach offers a novel direction for advancing expression recognition with current VLM.

**Future Direction: Self-Correction for Expression Recognition** Uni-MuMER provides a new perspective for enabling self-correction during inference in expression recognition. By integrating the Tree-CoT mechanism with an explicit error correction task, the model can exhibit self-corrective behavior within CoT process. Moreover, incorporating colloquial, human-like expressions (e.g., "wait, there's a left parenthesis earlier—this is more likely a right parenthesis, not '2'") can further induce R1-like spontaneous self-correction [19], enhancing the upper limit of the model's performance.

**Limitations** Despite the strong results, our method has several limitations. Due to computational resource constraints, a 400k synthetic subset of the MathWriting dataset was unused, potentially limiting the diversity benefits. Additionally, exploring optimal task mixture ratios within Uni-MuMER and optimal domain mixture ratios between HMER-specific data and general-domain data is crucial; however, these experiments were omitted due to significant resource constraints. Investigating these ratios could yield further enhance performance and generalizability.

## 7 Conclusion

This paper introduces Uni-MuMER, a unified multi-task fine-tuning framework for HMER , establishing a new paradigm by leveraging large-scale VLMs integrating multiple HMER tasks in a unified textual modality. Without architectural modifications, Uni-MuMER integrates Tree-aware Chain-of-Thought, Error-Driven Learning, and Symbol Counting tasks to jointly address structured spatial reasoning, error correction, and long-expression consistency in expression recognition, achieving super SOTA performance. This unified method provides enhanced accuracy, scalability, and faster inference speed, establishing a promising new research direction for future advancements in HMER.

# 8 Acknowledgement

This work is supported by the projects of Beijing Nova Interdisciplinary Program (20240484647) and National Natural Science Foundation of China (No. 62376012), which is also a research achievement of State Key Laboratory of Multimedia Information Processing and Key Laboratory of Science, Technology and Standard in Press Industry (Key Laboratory of Intelligent Press Media Technology).

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

# Appendix

## Table of Contents

# A   Prompt Design for Multi-Task Training

We design distinct prompts for each of the four training tasks in Uni-MuMER. Each prompt guides the model on a specific sub-task, and the model's expected output format is tailored to that task. Each prompt consists of a fixed system prompt: *You are a helpful assistant.* Below we list the prompts used for Vanilla HMER, Tree-CoT, Error-Driven Learning, and Symbol Counting (SC), along with example prompts and the form of expected outputs. It is a detailed expansion of the content in Fig. 2.

---

**Example A.1: Prompt for Vanilla HMER**

**Prompt:** I have an image of a handwritten mathematical expression. Please write out the expression of the formula in the image using LaTeX format.

**Answer:**

```
| \frac { a x _ { 0 } + b y _ { 0 } + c } { \sqrt { a ^ { 2 } + b ^ { 2 } } } |
```

---

**Example A.2: Prompt for Tree-CoT**

**Prompt:** I have an image of a handwritten mathematical expression. Please generate the **abstract syntax tree (AST)** of the formula in the image, and then provide its corresponding LaTeX format.

**Answer:**

```
|
\frac (right)
  a (above)
     x (right)
          0 (sub)
     + (right)
     b (right)
     y (right)
          0 (sub)
     + (right)
     c (right)
     \sqrt (below)
          a (above)
               2 (sup)
          + (right)
          b (right)
               2 (sup)
| (right)
\frac { a x _ { 0 } + b y _ { 0 } + c } { \sqrt { a ^ { 2 } + b ^ { 2 } } } |
```

---

**Example A.3: Prompt for Error Detection**

**Prompt:** I have an image of a handwritten mathematical expression and its OCR recognition result. Please help me to **detect** possible errors in the recognition result and mark the places where errors occur with <error_start> <error_end> and <deleted>.
erroneous formula: |\frac { a x _ { 0 } + b y _ { 0 } + c } { \sqrt { d ^ { 2 } + b ^ { 2 } } } |
Marked formula:

**Answer:**
|\frac { a x _ { 0 } + b y _ { 0 } + c } { \sqrt { <error_start> d <error_end> ^ { 2 } + b ^ { 2 } } } |

---

**Example A.4: Prompt for Error Correction**

**Prompt:** I have an image of a handwritten mathematical expression and a predicted formula with error tags, **correct** the formula by modifying the parts marked with <error_start> and <error_end> and inserting content where <deleted> are present. Output the modifications in a single string using the following format: REPLACE:old -> new for errors to be replaced. INSERT:content after the position for missing content. DELETE:to_delete for parts to be removed.
Marked formula: |\frac { a x _ { 0 } + b y _ { 0 } + c } { \sqrt { <error_start> d <error_end> ^ { 2 } + b ^ { 2 } } } |
Correction log:

**Answer:**
REPLACE:d -> a
|\frac { a x _ { 0 } + b y _ { 0 } + c } { \sqrt { a ^ { 2 } + b ^ { 2 } } } |

These carefully designed prompts ensure that during fine-tuning, Uni-MuMER learns not only to recognize the expressions but also to reason about structure, identify and correct errors, and pay attention to every symbol. By training on this multi-task prompt set, the model acquires a richer understanding of handwritten math, which cumulatively improves its accuracy and robustness.

# B  Expreimental Details

## B.1  Uni-MuMER-Data Preprocess

We perform comprehensive preprocessing on all datasets to prepare the training and evaluation data for Uni-MuMER. This involves cleaning noisy data, tokenizing LATEX strings into standardized tokens, and normalizing different LATEX notations to a consistent style. LATEX expressions can often be written in syntactically different ways while conveying the same meaning, so unifying these representations is crucial for effective cross-dataset training.

**Data Cleaning**  First, we exclude all instances containing CJK characters. For example, the HME100K dataset occasionally contains Chinese characters in annotations; these were removed to avoid confusing the model. Additionally, we remove characters or symbols that lack semantic value and negatively impact recognition accuracy, such as \underline{\quad} and empty brace pairs ({}), which do not contribute to the math content.

**Data Tokenization**  Since the LATEX expressions provided by datasets like CROHME 2023 [56] and Mathwriting [17] are not inherently tokenized. To train a sequence model effectively, we convert each expression into a standardized sequence of LATEX tokens. Specifically, we adopt the tokenization procedure from the `preprocess_formulas` script of the open-source `im2markup` repository, leveraging the KaTeX JavaScript library to perform semantic-level tokenization. During tokenization, we also remove any syntactically invalid expressions that fail to parse. Finally, the expression `\frac a^2 2` is tokenized as `\frac a ^ 2 2`, where each LATEX token is separated by a space.

**Data Normalization**  After tokenization, we normalize the ground-truth LATEX sequences across all training datasets to ensure a consistent format. Different datasets might represent the same mathematical concept in slightly different LATEX forms. Such variations include notation differences (e.g., using \leq and \le), optional braces (a^ 2 and a ^ { 2 } ), or stylistic markup (e.g., using \textbf{a} vs. just a), which is typically not visible in handwritten notes. If left unnormalized, these variations could confuse the model or reduce its ability to generalize between datasets. We therefore systematically unify these to a single canonical form. For example, the expression `\frac a^ 2 2` is normalized as `\frac { a ^ { 2 } } { 2 }`. Detailed normalization transformations are summarized

Table 8: **Examples of LATEX math normalization.** Comparison of original math expressions, normalized versions, and their rendered figures. Following the configurations of HiE-VL [21] and MathNet [44], we apply these transformations to make training more consistent.

| Transform | Original | Normalized | Figure |
|-----------|----------|------------|--------|
| Brace | H^I | H^{I} | $H^I$ |
| SubSup | \int^{b}_{a} | \int_{a}^{b} | $\int_a^b$ |
| Root | \sqrt[2]a | \sqrt[2]{a} | $\sqrt[2]{a}$ |
| Stylized | \textbf{a} | a | a |
| $\cdots$ | $\cdots$ | $\cdots$ | $\cdots$ |

in Tab. 8, illustrating the specific LATEX syntax standardization rules we applied, extending and optimizing the practices initially established by MathNet [44].

Table 9: **Dataset filtering summary and multi-task sample counts.**

| Category | CROHME | CROHME2023 | HME100K | Mathwriting | Im2LaTeXV2 |
|----------|--------|------------|---------|-------------|------------|
| raw | 8834 | 12204 | 74502 | 229864 | 74244 |
| rendering errors | 0 | 587 | 1166 | 624 | 1153 |
| normalization step | 0 | 1 | 590 | 4749 | 3954 |
| brackets invalid | 0 | 0 | 0 | 65 | 82 |
| rendering errors | 0 | 0 | 0 | 293 | 1 |
| Total (after deletion) | 8834 | 11616 | 72746 | 224133 | 69054 |
| Tree-CoT | 8834 | 11616 | 72746 | 224133 | 69054 |
| Error Detection | 9010 | 6287 | 49959 | 125316 | 8828 |
| Error Correction | 11434 | 8860 | 58884 | 166387 | 17047 |
| Symbol Counting | 8834 | 11616 | 72746 | 224133 | 69054 |
| Total (with four tasks) | 38112 | 38379 | 254335 | 739969 | 163983 |

**Dataset Filtering** Each dataset contributed a different number of valid expressions after preprocessing. Tab. 9 summarizes the filtering outcomes for the major datasets we used: CROHME, CROHME2023, HME100K, MathWriting, and Im2LaTeXv2. For each dataset, we list the original number of expressions and how many were removed for various reasons, such as rendering errors or invalid syntax detected during normalization. The final row gives the count of expressions remaining in each dataset after all filtering steps.

**Task Data Construction** With the cleaned datasets in hand, we construct the multi-task training samples. Each image can yield multiple training samples, one for each task. The lower part of Tab. 9 lists the number of training samples per task for each dataset. For the Tree-CoT, Symbol Counting, and Vanilla HMER tasks, we typically generate one sample per image, so their counts per dataset are equal to the number of remaining images. For the Error Detection and Error Correction subtasks, we collect all error corpus generated by each dataset. Further inspired by recent work on incremental refinement LATTE [26], our error-correction task is specifically constrained to correcting one error per round, which is more effective than inputting the whole incorrect expression. In total, across all datasets, our training set comprises roughly 1.6 million samples derived from about 392k unique images, spanning the three tasks.

## B.2 Evaluation Datasets and Metrics

**Datasets** The **MNE (Multi-level Nested Expression)** dataset is specifically designed to evaluate models on recognizing complex handwritten mathematical expressions [18]. The dataset is organized into subsets designated as N1, N2, and N3, based on the structural complexity of the expressions, determined by the maximum nested levels within their substructures. Fig. 7 illustrates various mathematical expressions along with their corresponding tree-based structural complexities. For instance, an expression such as $\frac{a^2+1}{b}$ possesses a structural complexity level of 2, indicating two hierarchical nesting layers.

**Evalution Metric** The **Character Error Rate (CER)** CER is computed via Levenshtein distance at the Unicode character level, normalized by ground-truth length. It is widely used in online handwriting datasets (e.g., *MathWriting*) to detect localization errors, brace imbalances, and missing superscripts. **EditScore** Following *MathNet*, we adopt EditScore, defined as $1 - \frac{\text{Edit distance}}{\max(|\hat{y}|,|y|)}$. EditScore, the sequence-level counterpart to CER, strongly correlates with human judgments in printed Mathematical Expression Recognition (MER) tasks and tolerates minor reordering and length variations while preserving semantics. **BLEU-4** For consistency and comparability with existing corpus in the im2latexv2 domain, we incorporate the BLEU-4 metric, calculated based on 4-gram overlap after LaTeX normalization, which effectively quantifies n-gram overlaps and remains a valuable metric.

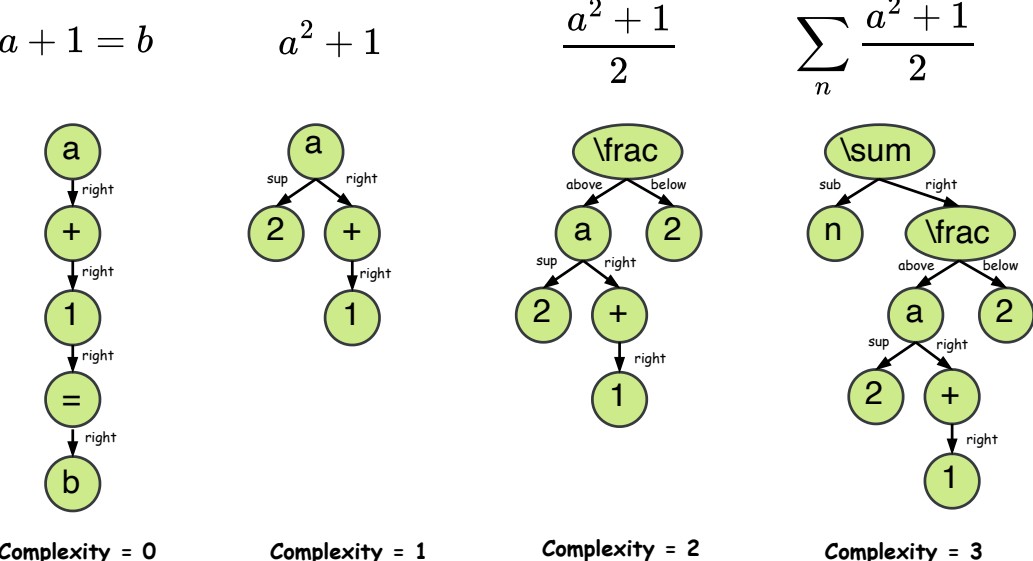

Figure 7: Examples of structural complexity for different expressions

## B.3 Evaluation Prompting Protocol

When evaluating the fine-tuned Uni-MuMER model and comparing it to various baselines, we standardize the prompting method to ensure a fair and consistent evaluation across models: Below, we describe the exact prompts and settings used in different evaluation scenarios.

**Uni-MuMER (Ours)** We used the Vanilla HMER prompt to elicit the final LaTeX output when evaluating our fine-tuned model on the test sets. Instead of explicitly prompting the model to show its chain-of-thought or to count symbols during this evaluation phase, we directly request the final LaTeX output. The rationale is that our multi-task training has already imbued the model with better internal reasoning and understanding; at test time, we just want the final answer, not the intermediate steps. Thus, for each test image, we simply provided the vanilla prompt like: "*I have an image of a handwritten mathematical expression. Please write out the expression of the formula in the image using LaTeX format.*" to the model, along with the handwritten expression image.

**Baseline VLMs (Open-Source and Closed-Source)** We evaluate several open-source VLMs and closed-source VLMs in a zero-shot setting on the CROHME and HME100K datasets. However, one challenge was that some general models might produce additional explanatory text or not output the expression alone. To mitigate this, we minimally revise the prompt to ask the model to output the LaTeX expression in the \boxed{}. The prompt is "*I have an image of a handwritten mathematical expression. Please write out the expression of the formula in the image using LaTeX format and place it inside \boxed{}*". We then post-processed the output by removing the \boxed{} wrapper to retrieve the raw LaTeX.

**Output Normalization** After obtaining outputs from all VLMs in a zero-shot setting, we apply the same cleaning and tokenization steps as used in training to each output before evaluation. This is important so that metrics like ExpRate or CER treat all outputs on equal footing. In particular, for the HME100K test set, if a model produced any non-LaTeX characters or CJK text, we dropped those outputs entirely.

**Reproducibility of Baselines** To ensure transparency and reproducibility, we explicitly detail the exact versions of the models evaluated in this study. Closed-source models were accessed through their respective APIs or publicly available releases at specific reference dates: Gemini 2.5-flash (version: *gemini-2.5-flash-preview-04-17* ), Doubao-1.5-pro (version: *doubao-1.5-vision-pro-250328* ), and GPT-4o (version: *gpt-4o-2024-11-20* ). For open-source models such as Qwen2.5-VL (3B, 7B, and 72B), we adopted the latest publicly accessible checkpoints available at the time of experimentation. All models underwent evaluation on identical test datasets, specifically the CROHME 2014, 2016, and 2019 datasets, as well as the HME100K test set. We meticulously ensured consistent preprocessing procedures and uniformly applied evaluation metrics across all model outputs.

By enforcing this consistent prompting and processing protocol, we aimed to make the comparison as fair as possible. All outputs were judged by exactly the same criteria. This way, the results reflect the true capability of each model on HMER, without artifacts from prompting differences.

## B.4 Training Detail

**Implementation Details** We fine-tune Qwen2.5-VL-3B using a full-parameter update. Besides, we employ a cosine decay schedule to gradually reduce the learning rate to 10% of its initial value $(1 \times 10^{-5})$ by the end of training. Training is conducted for a single epoch using all tasks and mixed dataset batches. Due to memory constraints, we adopt mixed-precision training, which enables us to fit a batch size of 512 within approximately 80 GB of VRAM per GPU (8×A800). This large batch size was specifically chosen to accommodate the diversity of tasks and stabilize the training process.

**Training time** We trained for a single epoch over the combined multi-task dataset. One epoch in this context amounted to roughly 1.6 million training samples, since each image contributes multiple task-specific samples. Iterating over 1.6M samples with batch 512 yields about 3152 optimization steps. In practice, this single epoch took 20 hours to complete on the $8\times$ A800 setup. Although this runtime slightly exceeds that of training lightweight specialized models from scratch, our approach achieves convergence to a high-performance level within just a single epoch.

## C  Extended Analysis

### C.1  Effects of Data Diversity Compared with Specialized HMER Model

In this subsection, we examine how incrementally scaling training data impacts ExpRate@CDM. As noted in Sec. 1, previous specialized models typically train on individual datasets, limited by the available training data and rarely assessed using visual metrics. Building upon the analysis in Sec. 5.2, we start with the CROHME dataset alone and then progressively add more training data from four additional sources. At each stage, we train three models from scratch on the combined data up to that point: (i) a lightweight specialized model, CoMER, which is a strong transformer-based architecture representative of specialized models; (ii) our vanilla baseline, which is only fine-tuned on 392k unique images; (iii) Uni-MuMER which is multi-task fine-tuned on the 1.6M samples).

The results, summarized in the Tab. 10, indicate that CoMER initially benefits from increased data diversity but subsequently exhibits significant performance degradation, settling at modest performance. Conversely, the vanilla baseline and Uni-MuMER show a much more robust improvement curve. Finally, Uni-MuMER consistently outperforms the Vanilla model at every stage and shows continuous gains. This demonstrates that the multi-task fine-tuning is effective at harnessing diverse data. Furthermore, specialized models like CoMER typically require extensive hyperparameter tuning, including adjustments to learning rates and training epochs. In contrast, Uni-MuMER attains superior performance after only a single training epoch without additional optimization efforts. This underscores Uni-MuMER's robustness and efficiency.

Table 10: **Effect of progressively adding training datasets on CROHME-Avg ExpRate@CDM**. At each step, models are trained from scratch on the cumulative data. Uni-MuMER improves monotonically and consistently outperforms the Vanilla (single-task) and CoMER baselines.

| Model | CROHME Train | + CROHME-23 | + HME100K | + MathWriting | + Im2Latexv2 |
|---|---|---|---|---|---|
| CoMER | 58.14 | 60.08 | 63.10 | 52.17 | 52.29 |
| Vanilla (baseline) | 71.14 | 75.07 | 75.13 | 80.87 | 80.35 |
| Uni-MuMER | 75.36 | 76.27 | 78.01 | 83.20 | 82.89 |
| Uni-MuMER-7B | 77.45 | 81.17 | 80.35 | 83.35 | 83.34 |

### C.2  Inference Efficiency

We evaluated the inference speed and model size of our proposed Uni-MuMER model against the baseline methods CoMER and TAMER, utilizing a single NVIDIA A800 GPU. To ensure a consistent and fair comparison, we used the HME100K test dataset, comprising 24,607 real-scene handwritten mathematical expression images. Additionally, we leveraged vLLM, an optimized transformer-based inference engine known for accelerating decoding processes through efficient batch processing and caching mechanisms. The comparative efficiency metrics are summarized in Table 11. As demonstrated, Uni-MuMER

Table 11: **Comparison of methods on efficiency metrics.** Throughput is reported in frames per second (FPS, higher is better) and model capacity in trainable parameters.

| Method | FPS ↑ | Parameters |
|---|---|---|
| CoMER | 5.23 | 6.39 M |
| TAMER | 4.50 | 8.23 M |
| **Uni-MuMER**[†] | 43.40 | 3.75 B |

significantly surpasses previous HMER models in terms of inference speed, achieving an FPS approximately eightfold higher than existing baselines under identical hardware conditions. As a result, Uni-MuMER is not only accurate but also efficient enough for practical use.

### C.3  Alternative explorations format

In developing the Tree-CoT and Symbol Counting tasks, we explored various output representation formats and their impact on model performance. Our goal was to find a format that is expressive enough to capture structure/counts but concise enough for the model to learn effectively. Here we report experiments with alternative formats for the expression tree serialization in Tree-CoT, as well as for the symbol counting task. We compare each alternative to the format ultimately used in Uni-MuMER (referred to as "our format"). All experiments in this section were conducted using

only CROHME data for training (no external data), so that differences reflect format efficacy rather than additional data. We evaluate both in a Vanilla+X setting and in the full Uni-MuMER setting (all three tasks, but substituting the alternative format for the relevant task) to assess each format's influence. Below, we outline each representation with an example.

---

**Example C.1: S-expression foramt for Tree CoT**

\mu_{pj}

---

**Example C.2: Json AST foramt for Tree CoT**

['type': 'supsub', 'value': 'base': 'type': 'mathord', 'value': '\mu', 'sub': 'type': 'ordgroup', 'value': ['type': 'mathord', 'value': 'p', 'type': 'mathord', 'value': 'j']]

---

**Example C.3: Simplified Json AST foramt for Tree CoT**

[supsub: base: '\mu', sub: ordgroup: ['p', 'j']]

---

The S-expression is merely the formula itself and adds no metadata. Its nested structure also hurts readability and organization. The JSON AST is excessively complex—it reflects JavaScript's internal LaTeX format—and includes substantial redundancy. The "Simplified JSON AST" was manually trimmed and lacks a consistent standard. In contrast, our AST format (used in Uni-MuMER) strikes a balance between structural clarity and brevity with special delimiters to indicate hierarchical relationships. Our format avoids the heavy overhead of full JSON while preserving the necessary structure of the expression tree.

Tab. 12 reports the model performance using each of these tree formats, which shows that the choice of format significantly impacts accuracy. We attribute this gain to our format's compactness and structured guidance: it provides clear structural cues (like JSON) but with far fewer extraneous tokens, making it easier for the decoder to learn the task.

Table 12: **Performance impact of different expression tree formats for Tree-CoT. Bold** numbers are best in column. Setup: Models are trained on the CROHME dataset and evaluated on CROHME 2014, 2016, and 2019 test sets and their average. Results are reported in terms of ExpRate and ExpRate@CDM. We compare the baseline Vanilla model against variants that incorporate the Tree-CoT task using various AST output formats, and likewise compare full Uni-MuMER models using those formats.

| Model | CROHME 2014 | | CROHME 2016 | | CROHME 2019 | | CROHME Average | |
|---|---|---|---|---|---|---|---|---|
| | Exp | @CDM | Exp | @CDM | Exp | @CDM | Exp | @CDM |
| Vanilla | 72.62 | 75.66 | 64.26 | 67.83 | 69.14 | 71.98 | 68.67 | 71.82 |
| Vanilla + S-expression | 72.21 | 75.56 | 66.87 | 70.97 | 68.06 | 71.73 | 69.05 | 72.75 |
| Vanilla + Json AST | 71.70 | 75.66 | 66.70 | 70.51 | 70.56 | 74.04 | 69.65 | 73.40 |
| Vanilla + Simplified Json AST | 72.62 | 76.47 | 66.96 | 70.18 | 69.89 | 73.06 | 69.82 | 73.24 |
| Vanilla + our AST | 73.33 | 75.76 | 68.00 | 70.97 | 71.23 | 74.40 | 70.85 | 73.71 |
| Uni-MuMER (w/ S-expression) | 74.85 | 78.07 | 68.07 | 73.14 | 72.56 | 76.14 | 71.83 | 75.78 |
| Uni-MuMER (w/ Json AST) | 75.46 | 79.51 | 70.10 | 74.28 | 72.98 | 76.06 | 72.85 | 76.62 |
| Uni-MuMER (w/ Simplified Json AST) | 74.44 | 78.27 | 70.36 | 74.63 | 73.90 | 77.48 | 72.90 | 76.79 |
| Uni-MuMER (w/ our AST) | 75.36 | 79.11 | **70.79** | **75.24** | **73.73** | **77.23** | **73.29** | **77.19** |

In addition, we investigated alternative formats for the Symbol Counting (SC) auxiliary task. The default format in Uni-MuMER expresses the symbol counts as a simple list of symbol-count pairs, separated by commas. We compared this with a more explicit map/dictionary format. For example, consider the expression x ˆ { 2 } + y ˆ { 2 } < 1.

---

**Example C.4: Map Counting format for Symbol Counting**

```
{'x': 1, 'ˆ': 2, '2': 2, '+': 1, 'y': 1, '<': 1, '1': 1}
```

---

**Example C.5: Our Counting format for Symbol Counting**

```
x: 1, ˆ: 2, 2: 2, +: 1, y: 1, <: 1, 1: 1
```

---

The Map Counting format is semantically equivalent to ours but clearly more redundant (it introduces additional punctuation and quotation characters). We found that this extra overhead can make the sequence longer and potentially confuse the model (especially because symbols like braces and quotes have no direct meaning in the counting task itself). As a result, as shown in Tab. 13, the map format yielded slightly worse performance. The differences are modest, but consistent.

Table 13: **Impact of Symbol Counting formats. Bold** numbers are best in column. Setup: Models are trained on the CROHME dataset and evaluated on CROHME 2014, 2016, and 2019 test sets and their average. Results are reported in terms of ExpRate and ExpRate@CDM. We compare using a JSON-like Map Counting format versus our simple listing format for the SC task.

| Model | CROHME 2014 | | CROHME 2016 | | CROHME 2019 | | CROHME Average | |
|---|---|---|---|---|---|---|---|---|
| | Exp | @CDM | Exp | @CDM | Exp | @CDM | Exp | @CDM |
| Vanilla | 72.62 | 75.66 | 64.26 | 67.83 | 69.14 | 71.98 | 68.67 | 71.82 |
| Vanilla + Map Counting | **72.92** | 76.67 | 65.39 | 70.01 | 69.89 | 73.65 | 69.40 | 73.44 |
| Vanilla + our Counting | 71.91 | **76.85** | 67.04 | **72.28** | **70.64** | **75.15** | **69.86** | **74.76** |
| Uni-MuMER (w/ Map Counting) | 75.05 | 78.80 | 70.62 | 74.43 | 72.48 | 76.21 | 72.72 | 76.48 |
| Uni-MuMER (w/ our Counting) | **75.36** | **79.11** | **70.79** | **75.24** | **73.73** | **77.23** | **73.29** | **77.19** |

## C.4 Hallucination for HMER

Large generative models are known to occasionally hallucinate output tokens that do not belong or omit tokens that should be present. In the context of HMER, both big VLMs and prior specialized models can produce spurious symbols. To quantitatively analyze this, we preliminarily adopt two metrics: Hallucination Token Rate (HTR) (percentage of predicted tokens not in the ground truth) and Missing Token Rate (MTR) (percentage of ground truth tokens not in the pred). Additionally, we categorized hallucinations by token type (numerical(0-9), variable(a-zA-Z), structural delimiters (e.g., [], {}, ()). This gives finer-grained error counts: HTR_num, HTR_var, HTR_delim for hallucinated numbers, letters, and delimiters, respectively. We evaluated these metrics across the CROHME 2014, 2016, and 2019 test sets, ensuring that no external data was used during training. The results are summarized below:

Table 14: **Error analysis of hallucinations and omissions on CROHME.** Results include ExpRate (Exp), ExpRate@CDM, overall HTR and MTR errors, as well as specific error types (HTR_num, HTR_var, HTR_delim). We compare two previous SOTA models ( CoMER and TAMER ), the Vanilla baseline, ablations adding each of our tasks individually (Tree-CoT, EDL, SC), and the full Uni-MuMER.

| Model | Exp | @CDM | HTR | MTR | HTR_num | HTR_var | HTR_delim |
|---|---|---|---|---|---|---|---|
| CoMER (SOTA'2022) | 59.18 | 62.47 | 2.55 | 3.91 | 0.37 | 0.33 | 1.18 |
| TAMER (SOTA'2024) | 60.29 | 63.18 | 2.17 | 4.02 | 0.29 | 0.29 | 1.05 |
| Vanilla (baseline) | 68.64 | 71.74 | 4.08 | 2.42 | 1.08 | 0.29 | 1.68 |
| Vanilla + Tree-CoT | **70.85** | 73.71 | **1.67** | **1.56** | 0.14 | **0.21** | 0.85 |
| Vanilla + EDL | 70.30 | 73.76 | 2.58 | 1.94 | 1.09 | 0.22 | **0.83** |
| Vanilla + CAN | 69.86 | **74.76** | 2.15 | 2.06 | 0.23 | 0.27 | 1.08 |
| Uni-MuMER (w/ 3 tasks) | **73.29** | **77.19** | **1.18** | **1.39** | **0.12** | **0.15** | **0.60** |

Tab. 14 shows that our full model yields the highest accuracy while also having the lowest HTR and MTR of all methods. Besides, incorporating each auxiliary task individually already helps: adding the Tree-CoT task slashes hallucinations to 1.67% HTR, adding the Symbol Counting task alone significantly lowers omissions (MTR), and adding the Error-Driven Learning task reduces hallucinations of delim and var. These improvements suggest that Tree-CoT primarily mitigates structural hallucinations, EDL corrects symbol-level mistakes, and SC enforces global consistency.

## C.5 CJK influence for HMER

In Sec. 4.2, we excluded samples containing CJK characters of HME100K. This is primarily for two reasons:

1. Our prompt specifically targets recognizing mathematical expressions, not general textual content such as CJK characters. This filtering ensures a clear evaluation of expression recognition capability.

2. Some CJK-containing samples introduced problematic annotations unrelated to pure expressions (e.g., textual descriptions or numeric lists)

Table 15: **Examples of CJK-containing image–caption pairs in HME100K that lie outside pure expression recognition**, which influence the robustness of expression recognition and are outside the scope of math expression recognition.

| image id | caption |
|---|---|
| train_63827 | = 88、5 |
| train_11042 | 1、3、14 × (9.42 ÷ 3.14) |
| train_70305 | 39的因数: 1, 3, |

Having justified the filtering strategy, we additionally examine the effect of **CJK** characters. We first list typical CJK-containing captions to illustrate the distributional shift (Tab. 15), then compare performance under two evaluation settings: (i) test set filtered to remove CJK, and (ii) test set retaining CJK. We also test training variants that include or exclude CJK samples. For reference, we also include results for CoMER and TAMER (previous specialized models) evaluated on the same splits.

Table 16: **Comparison of recognition performance on HME100K test sets with and without CJK characters.** Results are reported in terms of ExpRate and ExpRate@CDM. We evaluate three models on each subset: CoMER (specialized, SOTA 2022), TAMER (specialized, SOTA 2024), and Uni-MuMER. For Uni-MuMER, we report two training conditions: w/o CJK and w/ CJK. For CoMER and TAMER, we evaluated open-sourced models under zero-shot inference settings.

| Model | HME100K Test w/ CJK | | | HME100K Test w/o CJK | | |
|---|---|---|---|---|---|---|
| | ExpRate | ExpRate@CDM | CDM | ExpRate | ExpRate@CDM | CDM |
| CoMER | 68.47 | 70.19 | 96.79 | 68.70 | 70.29 | 96.81 |
| TAMER | 69.50 | 71.12 | 97.02 | 69.74 | 71.25 | 97.03 |
| Uni-MuMER[†] (w/o CJK) | 69.46 | 72.49 | 97.08 | **72.66** | 74.30 | 97.38 |
| Uni-MuMER[†] (w/ CJK) | **71.63** | **73.60** | **97.32** | 70.98 | **74.32** | 97.38 |

As is shown in Tab. 16, the performance of CoMER and TAMER remained nearly identical before and after filtering. Our model still outperforms previous methods, with the CJK characters the same in the training set, which suggests minimal effect on comparability, and including vs. excluding the CJK samples in training had minimal effect on overall HMER accuracy.

We further examine whether including CJK characters in training has any noticeable effect on the cross-domain performance of Uni-MuMER. Tab. 16 and Tab. 17 summarizes ExpRate@CDM results for two model variants across benchmarks. These results indicate that adding the small set of CJK-containing samples introduces a tiny trade-off: it may slightly compromise performance on the main handwriting benchmarks, while providing a negligible benefit to certain out-of-domain cases. We conclude that excluding the CJK data is a reasonable choice to keep the training focused, with no meaningful loss in generality.

Table 17: **Effect of including CJK during training evaluated on ExpRate@CDM across benchmarks.** Results are reported on CROHME-Average, CROHME 2023 test, Mathwriting test, and Im2Latexv2 test. The best results for each column are highlighted in bold.

| Model | CROHME-Avg | CROHME 2023 | Mathwriting | Im2Latexv2 |
|---|---|---|---|---|
| Uni-MuMER[†] (w/o CJK) | 82.89 | 74.42 | 69.11 | 88.99 |
| Uni-MuMER[†] (w/ CJK) | 82.21 | 72.14 | 68.91 | 90.28 |

# D  More Experimental Results

## D.1  Results on Handwritten Mathematical Expression

Tab. 1 and Tab. 2 in the main paper, we focused on the strict ExpRate and visual CDM metrics. Following past lightweight specialized models in HMER, we report $\leq 1$ and $\leq 2$ tolerating up to one or two token prediction errors within the generated LaTeX sequence. Here, we provide additional metrics and comparisons on multiple datasets to fully characterize Uni-MuMER's strengths.

Tab. 18 is an extended version of Tab. 1. Specifically, Tab. 18 reports not only ExpRate but also the lenient metrics allowing up to $\leq 1$ and $\leq 2$ token errors. We compare Uni-MuMER against a broad range of baselines: open-source VLMs, closed-source VLMs, previous SOTA methods on HMER, and their data augmentation versions.

Tab. 19 presents an extended evaluation of our method on the CROHME2023 test set, providing metrics including ExpRate, $\leq 1$ and $\leq 2$ token errors, as well as CDM-based evaluations. We benchmark Uni-MuMER and Uni-MuMER$^\dagger$ against various baselines. Our baseline model already demonstrates competitive performance against previous SOTA methods. Notably, after incorporating external data, Uni-MuMER$^\dagger$ exhibits a drop in ExpRate due to changes in LaTeX-style outputs but achieves outstanding performance in ExpRate@CDM.

Tab. 20 and Tab. 21 shows the results on the MNE dataset. This shows that the vanilla model is already competitive with the prior SOTA on these challenging sets. By introducing three data-driven tasks and external data, the Uni-MuMER and Uni-MuMER$^\dagger$ outperform the prior best methods.

Tab. 22 shows a comparison of MathWriting using CER. Mathwriting is a dataset of handwritten expressions paired with online handwriting traces and offline rendered images. With its introduction, Google released two foundation models, PaLI [14] and PaLM-E [13], which are capable of reading both images and sequences of pen strokes as input [14]. While online handwriting traces typically aid models in mitigating ambiguity among visually similar characters, resulting in enhanced performance, our proposed Uni-MuMER model demonstrates superior results despite relying exclusively on offline image data for both training and evaluation.

Tab. 23 is an extended version of Tab. 2, presenting the performance of various models on the HME100K test set. We report Exprate, $\leq 1$, $\leq 2$, and the CDM-based metrics for this dataset.

Table 18: **Performance comparison on CROHME datasets (%).** We evaluate our model and four baselines using expression recognition rate (Exp), $\leq 1$, and $\leq 2$ on the CROHME 2014, 2016, and 2019 test sets. We use the following notation for reported results: All results are zero-shot inference; [a]: data augmentation, [p]: reproduced results, $^\dagger$: external data used.

| Method | CROHME 2014 | | | CROHME 2016 | | | CROHME 2019 | | |
|---|---|---|---|---|---|---|---|---|---|
| | Exp↑ | $\leq 1$↑ | $\leq 2$↑ | Exp↑ | $\leq 1$↑ | $\leq 2$↑ | Exp↑ | $\leq 1$↑ | $\leq 2$↑ |
| GPT-4o | 50.61 | 65.21 | 77.69 | 46.03 | 61.99 | 73.85 | 49.79 | 64.79 | 75.06 |
| Doubao-1.5-pro | 50.41 | 68.15 | 77.69 | 49.00 | 61.99 | 73.85 | 46.87 | 68.81 | 78.65 |
| Gemini2.5-flash | 58.01 | 71.60 | 81.14 | 52.74 | 667.28 | 76.24 | 55.21 | 70.14 | 79.48 |
| Qwen2.5-VL-3B | 38.64 | 50.81 | 60.55 | 37.75 | 51.00 | 61.38 | 37.45 | 53.88 | 63.39 |
| Qwen2.5-VL-7B | 55.98 | 68.76 | 77.28 | 50.92 | 67.13 | 76.37 | 49.62 | 67.22 | 76.73 |
| Qwen2.5-VL-72B | 59.74 | 70.39 | 78.40 | 54.32 | 69.53 | 77.51 | 55.13 | 69.81 | 78.65 |
| DenseWAP | 50.1 | – | – | 47.5 | – | – | – | – | – |
| BTTR | 53.96 | 66.02 | 70.28 | 52.31 | 63.90 | 68.61 | 52.96 | 65.97 | 69.14 |
| ABM | 56.85 | 73.73 | 80.61 | 52.92 | 69.66 | 78.73 | 53.96 | 71.06 | 78.65 |
| CAN-ABM | 57.26 | 74.52 | 82.03 | 56.15 | 72.71 | 80.30 | 55.96 | 72.73 | 80.57 |
| CoMER | 58.38 | 74.48 | 81.14 | 56.98 | 74.44 | 81.87 | 59.12 | 77.45 | 83.87 |
| PosFormer | 60.45 | 77.28 | 83.68 | 60.94 | 76.72 | 83.87 | 62.22 | 79.40 | 86.57 |
| TAMER | 61.36 | 76.77 | 83.25 | 60.26 | 76.91 | 84.05 | 61.97 | 78.97 | 85.80 |
| SSAN | 60.85 | 75.56 | 82.25 | 60.02 | 76.22 | 83.28 | 61.83 | 79.08 | 86.08 |
| CoMER[a] | 59.33 | 71.70 | 75.66 | 59.81 | 74.37 | 80.30 | 62.97 | 77.40 | 81.40 |
| CoMER[ap†] | 46.96 | 60.65 | 69.27 | 43.33 | 56.84 | 64.60 | 48.29 | 64.22 | 71.64 |
| PosFormer[a] | 62.68 | 79.01 | 84.69 | 61.03 | 77.86 | 84.74 | 64.97 | 82.49 | 87.24 |
| SSAN[a†] | 62.58 | – | – | 62.51 | – | – | 65.30 | – | – |
| VLPG$^\dagger$ | 60.41 | 74,14 | 78.90 | 60.51 | 75.50 | 79.86 | 62.34 | 76.81 | 81.15 |
| HiE-VL[a†] | 73.30 | 84.00 | 87.00 | 70.70 | 82.10 | 86.40 | 69.30 | 81.90 | 86.20 |
| **Vanilla (baseline)** | 72.62 | 81.34 | 86.82 | 64.26 | 77.77 | 84.22 | 69.06 | 82.07 | 88.07 |
| **Uni-MuMER** | **75.36**$^{+2.74}$ | **85.09**$^{+3.75}$ | **89.05**$^{+2.23}$ | **70.79**$^{+6.53}$ | **83.44**$^{+5.67}$ | **89.54**$^{+5.32}$ | **73.73**$^{+4.67}$ | **86.74**$^{+4.67}$ | **91.83**$^{+3.76}$ |
| **Uni-MuMER$^\dagger$** | **82.05**$^{+9.43}$ | **89.45**$^{+8.11}$ | **93.00**$^{+6.18}$ | **77.94**$^{+13.68}$ | **87.45**$^{+9.68}$ | **92.07**$^{+7.85}$ | **79.23**$^{+10.17}$ | **90.91**$^{+8.84}$ | **93.83**$^{+5.76}$ |

Table 19: **Performance on CROHME2023 Test set (%).** Evaluated by ExpRate, ExpRate@CDM, and CDM. The $^{\dagger}$ denotes the use of external data. The $^{p}$ denotes the results reproduced by [55].

| Method | CROHME 2023 | | | | |
| --- | --- | --- | --- | --- | --- |
| | ExpRate ↑ | ≤ 1 ↑ | ≤ 2 ↑ | ExpRate@CDM ↑ | CDM ↑ |
| VLPG | 56.47 | 68.54 | 73.63 | – | – |
| CoMER $^{p}$ | 59.82 | 69.23 | 74.12 | – | – |
| TST | 60.72 | 70.51 | 75.00 | – | – |
| TST$^{\dagger}$ | 62.59 | 73.59 | 77.51 | – | – |
| Vanilla (baseline) | 68.39 | 79.13 | 87.57 | 69.24 | 94.43 |
| Uni-MuMER | 70.26$^{+1.87}$ | 81.13$^{+2.00}$ | 88.35$^{+0.78}$ | 71.47$^{+2.23}$ | 95.12$^{+0.69}$ |
| Uni-MuMER$^{\dagger}$ | 70.00$^{+1.61}$ | 79.78$^{+0.65}$ | 87.00$^{-0.57}$ | **74.42**$^{+5.18}$ | **95.33**$^{+0.90}$ |

Table 20: **Performance comparison on the MNE dataset (%).** We compare the expression recognition rate (ExpRate) and CDM-based metrics between our model and previous SOTA models on the MNE N1/N2/N3 test sets. The $^{\dagger}$ denotes the use of external data.

| Method | N1 | | | N2 | | | N3 | | |
| --- | --- | --- | --- | --- | --- | --- | --- | --- | --- |
| | Exp ↑ | @CDM ↑ | CDM ↑ | Exp ↑ | @CDM ↑ | CDM ↑ | Exp ↑ | @CDM ↑ | CDM ↑ |
| CoMER | 59.73 | – | – | 37.17 | – | – | 24.04 | – | – |
| PosFormer | 60.59 | – | – | 38.82 | – | – | 36.82 | – | – |
| MMHMER | 62.03 | – | – | 39.47 | – | – | 34.29 | – | – |
| Vanilla (baseline) | 64.37 | 68.21 | 93.94 | 51.65 | 58.22 | 92.89 | 35.66 | 46.59 | 91.75 |
| Uni-MuMER | 70.08$^{+5.71}$ | 74.19$^{+5.98}$ | 95.33$^{+1.39}$ | 55.59$^{+3.94}$ | 62.83$^{+4.61}$ | 94.64$^{+1.75}$ | 38.53$^{+2.87}$ | 50.96$^{+4.37}$ | 93.63$^{+1.88}$ |
| Uni-MuMER$^{\dagger}$ | **76.00**$^{+11.63}$ | **81.44**$^{+13.23}$ | **96.37**$^{+2.43}$ | **58.88**$^{+7.23}$ | **71.38**$^{+13.16}$ | **96.08**$^{+3.19}$ | **46.45**$^{+10.79}$ | **66.67**$^{+20.08}$ | **95.13**$^{+3.38}$ |

Table 21: **Performance comparison on the MNE dataset (%).** We compare the expression recognition rate (ExpRate), ≤ 1, and ≤ 2 between our model and previous SOTA models on the MNE N1/N2/N3 test sets. The $^{\dagger}$ denotes the use of external data.

| Method | N1 | | | N2 | | | N3 | | |
| --- | --- | --- | --- | --- | --- | --- | --- | --- | --- |
| | Exp ↑ | ≤ 1 ↑ | ≤ 2 ↑ | Exp ↑ | ≤ 1 ↑ | ≤ 2 ↑ | Exp ↑ | ≤ 1 ↑ | ≤ 2 ↑ |
| CoMER | 59.73 | 77.55 | 84.11 | 37.17 | 53.95 | 65.13 | 24.04 | 32.31 | 36.34 |
| PosFormer | 60.59 | 77.97 | 84.32 | 38.82 | 56.91 | 66.12 | 36.82 | 40.30 | 43.10 |
| MMHMER | 62.03 | 78.13 | 84.32 | 39.47 | 56.57 | 65.46 | 34.29 | 36.68 | 40.51 |
| Vanilla (baseline) | 64.37 | 78.61 | 85.65 | 51.65 | 59.54 | 69.08 | 35.66 | 44.81 | 52.12 |
| Uni-MuMER | 70.08$^{+5.71}$ | 83.73$^{+5.12}$ | 90.29$^{+4.64}$ | 55.59$^{+3.94}$ | 64.80$^{+5.26}$ | 73.36$^{+4.28}$ | 38.53$^{+2.87}$ | 49.18$^{+4.37}$ | 57.24$^{+5.12}$ |
| Uni-MuMER$^{\dagger}$ | **76.00**$^{+11.63}$ | **87.31**$^{+8.70}$ | **92.16**$^{+6.51}$ | **58.88**$^{+7.23}$ | **74.34**$^{+14.80}$ | **82.90**$^{+13.82}$ | **46.45**$^{+10.79}$ | **54.37**$^{+9.56}$ | **63.66**$^{+11.54}$ |

Table 22: **Comparison with multi-modal models PaLI and PaLM-E on MathWriting.** CER is the character error rate (lower is better). The $^{\dagger}$ denotes the use of external data.

| Model | Input | CER ↓ | Exp ↑ | @CDM ↑ | CDM ↑ |
| --- | --- | --- | --- | --- | --- |
| PaLI | Image | 8.07 | – | – | – |
| PaLI | Ink | 4.64 | – | – | – |
| PaLI | Ink + Image | 4.55 | – | – | – |
| PaLM-E | Image | 4.87 | – | – | – |
| PaLM-E | Ink | 6.46 | – | – | – |
| PaLM-E | Ink + Image | 4.22 | – | – | – |
| **Vanilla** | Image | 4.15 | 50.51 | 66.46 | 94.20 |
| **Uni-MuMER** | Image | 4.01 | 51.19 | 68.73 | 94.80 |
| **Uni-MuMER**$^{\dagger}$ | Image | **4.00** | **51.45** | **69.11** | **94.84** |

## D.2 Results on Printed Mathematical Expressions

An important aspect of our approach is whether a model fine-tuned primarily for handwritten expressions also performs well on printed mathematical expressions. To test this, we evaluate on the Im2LaTeXv2 dataset, a dataset of printed formulas rendered from LaTeX. Minor normalization adjustments were applied to align the dataset with our preprocessing standards. Specifically, we performed synonym replacements such as changing expressions from \le to \leq, inserted spaces to separate commands clearly (e.g., converting \begin{align} to \begin {align}), and standardized

Table 23: **Performance on HME100K dataset (%).** Evaluated by Exp, @CDM, and CDM. The † denotes the use of external data.

| Method | HME100K | | | | |
| --- | --- | --- | --- | --- | --- |
| | Exp ↑ | ≤ 1 ↑ | ≤ 2 ↑ | @CDM ↑ | CDM ↑ |
| GPT-4o | 22.96 | 37.32 | 47.40 | 27.02 | 83.13 |
| Gemini2.5-flash | 28.14 | 44.23 | 54.75 | 34.32 | 85.43 |
| Doubao1.5-pro | 43.90 | 64.52 | 74.09 | 55.08 | 91.28 |
| Qwen2.5-VL-3B | 44.42 | 61.47 | 70.38 | 47.41 | 89.96 |
| Qwen2.5-VL-7B | 54.57 | 71.05 | 78.44 | 58.84 | 92.90 |
| Qwen2.5-VL-72B | 49.59 | 66.21 | 73.79 | 55.09 | 93.08 |
| DenseWAP | 61.85 | 70.63 | 77.14 | – | – |
| CoMER | 68.12 | 84.20 | 89.71 | 70.36 | 96.83 |
| TAMER | 69.50 | 85.48 | 90.80 | 71.30 | 97.07 |
| HiE-VL† | 64.20 | – | – | – | – |
| Vanilla (baseline) | 70.15 | 85.81 | 91.25 | 71.80 | 96.97 |
| Uni-MuMER | $71.62^{+1.47}$ | $86.92^{+1.11}$ | $92.16^{+0.91}$ | $73.40^{+1.60}$ | $97.30^{+0.33}$ |
| Uni-MuMER† | $\mathbf{72.66}^{+2.51}$ | $\mathbf{87.67}^{+1.86}$ | $\mathbf{92.60}^{+1.35}$ | $\mathbf{74.30}^{+2.50}$ | $\mathbf{97.38}^{+0.41}$ |

notation by enclosing single-element subscripts like `a_i` within braces as `a_{i}`. Other than these normalization steps, no further modifications were made, and the mathematical semantics remained entirely unaffected. These normalization procedures are consistent with the preprocessing approach detailed in Appendix B. For transparency, we also include results from evaluations performed on the original, unnormalized test set. Such normalization adjustments were not applied to other test sets in our study, thus preserving fairness and comparability across datasets.

Tab. 24 summarizes these results using common metrics from the Im2LaTeXv2: Edit Score, BLEU-4, and ExpRate. We also include ExpRate@CDM for completeness, especially since our models output LaTeX that could be visually equivalent even if not exactly matching. Our Vanilla baseline performed slightly better than the previous SOTA. After completing three data-driven tasks, Uni-MuMER further reached an ExpRate of 88.52. We also check our Uni-MuMER†, which is primarily optimized for handwriting. Interestingly, its performance on purely printed data is a bit lower than Uni-MuMER, perhaps because training on such a broad mix causes a slight domain shift. Nonetheless, all our models outperform MathNet [44].

Table 24: **Comparison of model performance on the *im2latexv2* test set**. We compare our models to prior Im2LaTeX systems. Metrics shown are Edit Score, BLEU-4, ExpRate, and ExpRate@CDM. All metrics are percentages (%), and higher values ↑ indicate better performance. The lower half reports results on the original, unnormalized test set. Missing values indicated by '–'.

| Model | Train Dataset | Edit | Bleu-4 | ExpRate | ExpRate@CDM |
| --- | --- | --- | --- | --- | --- |
| WYGIWYS | im2latex-100k | 37.2 | 23.9 | 0.0 | – |
| I2l-strips | im2latex-140k | 75.9 | 65.9 | 10.3 | – |
| I2l-nopool | im2latex-140k | 76.0 | 66.4 | 10.4 | – |
| MathNet | im2latexv2 | 97.2 | 96.8 | 83.9 | – |
| Vanilla (baseline) | im2latexv2 | 99.12 | 98.44 | 88.08 | 93.19 |
| Uni-MuMER | im2latexv2 | 99.16 | 98.70 | **88.52** | **93.46** |
| Uni-MuMER† | Uni-MuMER-Data | **99.33** | **98.71** | 88.19 | 93.29 |
| *Evaluation on Unnormalized Test Set* | | | | | |
| Vanilla (baseline) | im2latexv2 | 98.71 | 97.59 | 79.85 | 93.19 |
| Uni-MuMER | im2latexv2 | 98.75 | 97.85 | 80.27 | 93.46 |
| Uni-MuMER† | Uni-MuMER-Data | 98.54 | 96.72 | 76.36 | 93.29 |

# E  Case Study

We present a qualitative comparison to illustrate how Uni-MuMER handles a challenging expression versus other models. As shown in Fig. 8, we compare the outputs of closed-source VLMs, a prior lightweight HMER model (CoMER), our fine-tuned baseline (vanilla), and our Uni-MuMER. The figure displays each model's predicted LaTeX, highlighting any incorrect tokens in red. These case studies reinforce our quantitative findings: Uni-MuMER is not only quantitatively better but qualitatively produces significantly cleaner and more complete outputs, especially on expressions that involve several levels of nesting and a variety of symbols.

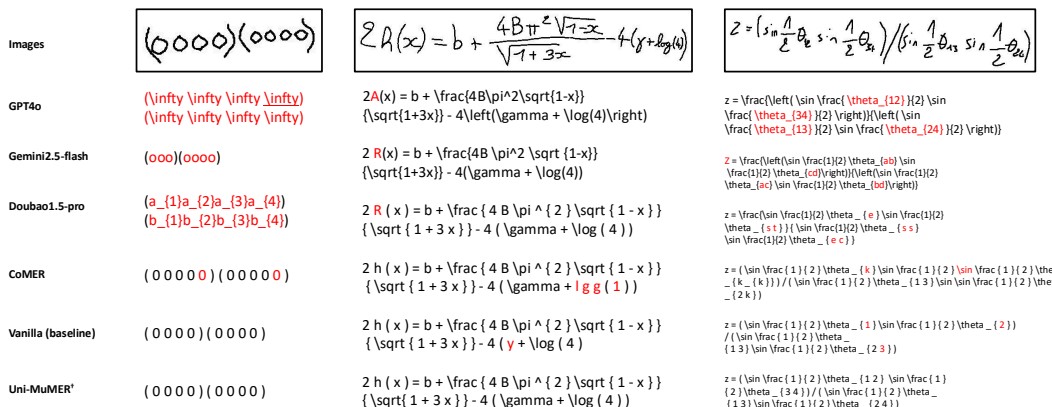

| | | | |
|---|---|---|---|
| **Images** | | | |
| **GPT4o** | (\infty \infty \infty \infty) (\infty \infty \infty \infty) | 2A(x) = b + \frac{4B\pi^2\sqrt{1-x}}{\sqrt{1+3x}} - 4\left(\gamma + \log(4)\right) | z = \frac{\left( \sin \frac{\theta_{12}}{2} \sin \frac{\theta_{34}}{2} \right)}{\left( \sin \frac{\theta_{13}}{2} \sin \frac{\theta_{24}}{2} \right)} |
| **Gemini2.5-flash** | (ooo)(oooo) | 2 R(x) = b + \frac{4B \pi^2 \sqrt{1-x}}{\sqrt{1+3x}} - 4(\gamma + \log(4)) | Z = \frac{\left(\sin \frac{1}{2} \theta_{ab} \sin \frac{1}{2} \theta_{cd}\right)}{\left(\sin \frac{1}{2} \theta_{ac} \sin \frac{1}{2} \theta_{bd}\right)} |
| **Doubao1.5-pro** | (a_{1}a_{2}a_{3}a_{4}) (b_{1}b_{2}b_{3}b_{4}) | 2 R ( x ) = b + \frac { 4 B \pi ^ { 2 } \sqrt { 1 - x } } { \sqrt { 1 + 3 x } } - 4 ( \gamma + \log ( 4 ) ) | z = \frac{\sin \frac{1}{2} \theta_{e} \sin \frac{1}{2} \theta_{st} \theta_{ss}}{\sin \frac{1}{2} \theta_{ec}} |
| **CoMER** | ( 0 0 0 0 0 )( 0 0 0 0 0 ) | 2 h ( x ) = b + \frac { 4 B \pi ^ { 2 } \sqrt { 1 - x } } { \sqrt { 1 + 3 x } } - 4 ( \gamma + l g g ( 1 ) ) | z = ( \sin \frac { 1 } { 2 } \theta _ { k } \sin \frac { 1 } { 2 } \sin \frac { 1 } { 2 } \theta _ { k _ { k } } ) / ( \sin \frac { 1 } { 2 } \theta _ { 1 3 } \sin \sin \frac { 1 } { 2 } \theta _ { 2 k } ) |
| **Vanilla (baseline)** | ( 0 0 0 0 )( 0 0 0 0 ) | 2 h ( x ) = b + \frac { 4 B \pi ^ { 2 } \sqrt { 1 - x } } { \sqrt { 1 + 3 x } } - 4 ( \gamma + \log ( 4 ) | z = ( \sin \frac { 1 } { 2 } \theta _ { 1 } \sin \frac { 1 } { 2 } \theta _ { 2 } ) / ( \sin \frac { 1 } { 2 } \theta _ { 1 3 } \sin \frac { 1 } { 2 } \theta _ { 2 3 } ) |
| **Uni-MuMER†** | ( 0 0 0 0 )( 0 0 0 0 ) | 2 h ( x ) = b + \frac { 4 B \pi ^ { 2 } \sqrt { 1 - x } } { \sqrt { 1 + 3 x } } - 4 ( \gamma + \log ( 4 ) ) | z = ( \sin \frac { 1 } { 2 } \theta _ { 1 2 } \sin \frac { 1 } { 2 } \theta _ { 3 4 } ) / ( \sin \frac { 1 } { 2 } \theta _ { 1 3 } \sin \frac { 1 } { 2 } \theta _ { 2 4 } ) |

Figure 8: **Qualitative comparison on a challenging handwritten expression.** Compared with closed-source VLMs, CoMER, Vanilla(baseline), and Uni-MuMER†. The red symbols represent incorrect predictions.

**CDM is not the ultimate evaluation metric yet** Although CDM metrics [50] quantify the visual equivalence between predicted and ground-truth LaTeX sequences beyond ExpRate, they are not definitive for evaluating LaTeX recognition accuracy. CDM tends to prioritize local character matching, potentially neglecting significant global structural discrepancies. This oversight can lead to inaccurate recognition assessments, as is shown in Fig. 9, CDM fails to detect the incorrect subscript structure of the character $u_x u_x$, mistakenly equating it with $uu_{xx}$, due to relying solely on local character matching without considering global hierarchical relationships.

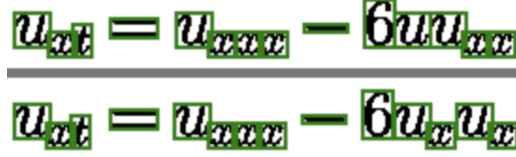

Figure 9: An example of CDM leading to incorrect recognition assessment.

