# OpenReview forum: "Uni-MuMER: Unified Multi-Task Fine-Tuning of Vision-Language Model for Handwritten Mathematical Expression Recognition"
_NeurIPS.cc/2025/Conference — NeurIPS 2025 spotlight_

### Official Review · Reviewer_Yrc6 · 2025-07-01

**Clarity:** 3
**Significance:** 2
**Originality:** 2
**Rating:** 4
**Confidence:** 5

**Summary:**

The paper proposes Uni-MuMER, a unified multi-task fine-tuning framework for Handwritten Mathematical Expression Recognition (HMER), leveraging a generalist vision-language model (Qwen2.5-VL-3B). The authors introduce three auxiliary tasks: Tree-Aware Chain-of-Thought (Tree-CoT) for structural reasoning, Error-Driven Learning (EDL) for self-correction, and Symbol Counting (SC) for improving recognition consistency. Experiments on the CROHME and HME100K datasets demonstrate strong improvements over existing models, including both zero-shot and fine-tuned baselines.

**Questions:**

1) In Table 1, several prior methods were trained solely on the CROHME training set. Could the authors report the performance of Uni-MuMER when fine-tuned only on the CROHME training set, without incorporating external data? This would help establish a fairer comparison under the same training conditions.

2) For the HME100K evaluation, the paper mentions excluding samples containing CJK characters. Could the authors clarify the rationale behind this filtering step, and whether it significantly affects the reported performance or comparability with prior work?

3) “External data” is mentioned in both Table 1 and Table 2. Could the authors clarify which datasets or sources are included in this category?

**Ethical Concerns:**

["NO or VERY MINOR ethics concerns only"]

**Final Justification:**

After the rebuttals, the authors have addressed and clairified most of my concerns.

**Limitations:**

Compared to specialized models, do large fine-tuned VLMs like Uni-MuMER exhibit hallucination or repetition issues when generating mathematical expressions? This is a common concern with large models and should be discussed in the limitations section.

**Quality:**

3

**Strengths And Weaknesses:**

Strengths:
1) The paper addresses a relevant and long-standing challenge in OCR for mathematical expressions, with a novel use of large-scale vision-language models.
2) The integration of three auxiliary tasks is well-motivated, and the experimental results show that they complement each other.
3) Ablation studies and comparisons with both open- and closed-source baselines are fairly comprehensive.

Weaknesses:
1) The method relies solely on fine-tuning and input reformulation without architectural innovation, which may limit its perceived novelty.
2) The Tree-CoT serialization and symbol counting formats are hand-designed and lack systematic justification or alternative explorations.
3) While the integration of multiple tasks into a unified framework is valuable, the individual tasks themselves (e.g., symbol counting and AST-based reasoning) are not new to the HMER domain. Prior works such as CAN and tree-decoder-based models have already explored similar directions.
4) The Tree-CoT task assumes expressions can be parsed into clean hierarchical trees, which limits its applicability to more complex layouts like matrices, piecewise functions, or multi-line expressions that follow grid-like or block structures rather than tree-like ones.

---

> ### Author Rebuttal · Authors · 2025-07-31
>
> # Response to Reviewer Yrc6
>
> We thank reviewer for thoughtful and constructive feedback. We address each concern and question in detail below.
>
> ## Weakness
>
> 1. **Lack of Architectural Innovation**
>
> > The method relies solely on fine-tuning and input reformulation without architectural innovation, potentially limiting perceived novelty.
>
> From past viewpoint with smaller models, framework innovation was essential; But for the **current generation** of **large-scale VLMs**, we argue that **data-centric and unified multi-task fine-tuning itself constitutes a significant innovation.**
>
> Although each innovation individually offers distinct advantages, as discussed in the paper (line 41), **prior architectural innovations are fragmented and difficult to integrate, which limits the development of HMER domain.**
>
> In contrast, **our Uni-MuMER leverages VLM input-output to integrate multiple architectural strengths, achieving SOTA performance**. This novel recognition paradigm can further generalize to related tasks and enhance robustness by integrating diverse data sources, providing new insights for applying VLMs in expression recognition tasks.
>
> Looking ahead, our approach extends in two promising directions:
>
> - enhancing general VLM capabilities through richer, domain-specific data resources;
> - integrating reinforcement learning strategies, such as R1-like self-reflective error correction,
>
> 2. **Auxiliary Tasks (AST Reasoning and Symbol Counting) Are Not New**
>
> > "...the tasks are not new to the HMER. Prior works such as CAN and tree-based models have explored similar directions."
>
> We appreciate the reviewer’s insightful comment.
>
> 1. Our contribution is not about proposing any single auxiliary task in isolation. Rather, we introduce a **unified modeling paradigm** capable of jointly addressing multiple HMER-related tasks (as detailed in our response to Weakness 1).
> 2. Specifically, the proposed Error Detection and Correction tasks represent a **novel direction** for HMER, laying the foundation for self-reflective capabilities in future HMER.
> 3. Previous works primarily experimented with smaller models. Their effectiveness when applied to modern large-scale pretrained VLMs has not been validated, and their improvements may not translate to current VLMs
> 4. Additionally, prior studies typically experimented on limited datasets. We found that the performance of these lightweight models often decreases significantly (as demonstrated in Appendix C.1 Table 7).
>
> Table 7: Impact of incrementally training data scaling on CROHME evaluaed with ExpRate@CDMmailto:ExpRate@CDM %
> ||CROHME |+ CROHME-23|+ HME100K|+ MathWriting|+ Im2Latexv2|
> |-|-|-|-|-|-|
> |CoMER|58.14|60.08|63.10|52.17|52.29|
> |Uni-MuMER|**75.36**|**76.27**|**78.01**|**83.20**|**82.86**|
>
>
> 3. **Alternative explorations format**
>
> > The Tree-CoT serialization and symbol counting formats are hand-designed and lack alternative explorations.
>
> For Tree-CoT serialization formats, including **S-expression**, **JSON AST** and our proposed **AST format**.
>
> - **S-expression** provides minimal structural detail.
>
>   - mu_{pj}
> - **JSON AST** is overly verbose and redundant.
>
>   - [{'type': 'supsub', 'value': {'base': {'type': 'mathord', 'value': '\\mu'}, 'sub': {'type': 'ordgroup', 'value': [{'type': 'mathord', 'value': 'p'}, {'type': 'mathord', 'value': 'j'}]}}}]
> - **our chosen AST** format offers a clear and concise hierarchical representation, balancing minimal redundancy with structural clarity.
>
> ```
> \mu
>     p (sub)
>     j (right)
> ```
>
> We evaluated our method across those formats, observing improved performance in all cases, with the our **AST format achieving the best results**
>
> ||14-Exp|14-CDM|16-Exp|16-CDM|19-Exp|19-CDM|Avg-Exp|Avg-CDM|
> |-|-|-|-|-|-|-|-|-|
> |Vanilla|72.62|75.66|64.26|67.83|69.14|71.98|68.67|71.82|
> |w/ S-exp|72.21|75.56|66.87|70.97|68.06|71.73|69.05|72.75|
> |w/ Json AST|71.70|75.66|66.70|70.51|70.56|**74.04**|69.65|73.40|
> |w/ Sim AST|72.62|**76.47**|66.96|70.18|69.89|73.06|69.82|73.24|
> |w/ our AST|73.33|75.76|68.00|70.97|71.23|74.40|70.85|73.71|
>
> We explored another counting format, “Map Counting” with higher redundancy. Experiments confirm our format are better.
>
> For `x + 1`
>
> **Map Counting Format**
>
> ```
> {'x':1,'+':1,'1' 1}
> ```
>
> **Our Counting Format**
>
> ```
> x: 1,+: 1,1: 1
> ```
>
> ||14Exp|14@CDM|16Exp|16@CDM|19Exp|19@CDM|AvgExp|Avg@CDM|
> |-|-|-|-|-|-|-|-|-|
> |Vanilla|72.62|75.66|64.26|67.83|69.14|71.98|68.67|71.82|
> |Van+Map|**72.92**|76.67|65.39|70.01|69.89|73.65|69.40|73.44|
> |Van+our|71.91|76.85|67.04|72.28|70.64|75.15|69.86|74.76|
>
>
>
> 4. **Applicability of Tree-CoT to Non-Tree Structures:**
>
> > The Tree-CoT task ... which limits its applicability to more complex layouts like matrices, piecewise functions, or multi-line expressions that follow grid-like or block structures rather than tree-like ones.
>
> We clarify that complex layouts **can indeed be transformed into hierarchical tree structures**, including these multi-line expressions. While earlier HMER datasets, such as CROHME, omitted these formats, recent datasets, such as **Mathwriting include them**. We address this gap by introducing relation tokens (e.g., nextline).
>
> For instance, a piece wise function:
>
> $$
> y = \begin{cases}  0 & x < 0 \\\\  1 & x \ge 0  \end{cases}
> $$
>
> ```
> y
> = (right)
> \begin (right)
>     0 (begin_cases)
>     & (right)
>     x (right)
>     < (right)
>     0 (right)
>           1 (nextline)
>           & (right)
>           x (right)
>           \ge (right)
>           0 (right)
>     \end (end_cases)
> ```
>
>
> ## Questions
>
> 1. **CROHME-Only Fine-Tuning Results**
>
> > Could you provide Uni-MuMER’s results when fine-tuned only on CROHME (without external data) for fair comparison with prior methods (Table 1)?
>
> In fact, We stated in Section 4.1 (lines 240), and our model surpassing previous SOTA.
>
> ||Score|
> |-|-|
> |SSAN(SOTA'25)|63.43|
> |Uni-MuMER|**73.29**|
>
> This setup applies consistently across Tables 2, 9–12. Both Vanilla and Uni-MuMER **were trained solely on the respective training sets**. We only employed external data in models explicitly marked with a dagger ($\dagger$), such as Uni-MuMER$^\dagger$.
>
>
> 2. **Filtering out CJK in HME100K Evaluation**
>
> > For HME100K, Could the authors clarify the rationale behind filtering CJK characters, and whether it significantly affects the reported performance or comparability with prior work?
>
> Thank you for raising this point. We excluded samples containing CJK characters primarily for two reasons:
>
> 1. Our prompt **specifically targets recognizing mathematical expressions**, not general textual content such as CJK characters. This filtering ensures a clear evaluation of expression recognition capability.
> 2. Some CJK-containing samples introduced problematic annotations unrelated to pure expressions (e.g., textual descriptions or numeric lists)
>
> |image_id|caption|
> |-|-|
> |train_63827|=88、5|
> |train_70305|39的因数：1,3,|
>
> We evaluated the **impact of this filtering and found minimal effect on comparability**.
>
> To further address your concern, we **retrained Uni-MuMER† with the CJK characters in training set** and compared its performance with w/o CJK version, evaluate on ExpRate (%):
>
> - For instance, the performance of CoMER and TAMER remained nearly identical before and after filtering.
> - **our model still outperforms previous methods**, with the CJK characters same in training set
>
> ||w/CJK|w/o CJK|
> |-|-|-|
> |CoMER|68.47|68.70|
> |TAMER|69.50|69.74|
> |Uni-MuMER⁺(w/o CJK)|69.46|72.72|
> |Uni-MuMER⁺(w/CJK)|**71.63**|70.98|
>
> However, **a decrease in expression recognition performance when add CJK into training set.**
>
> ||CROHME-Avg|
> |-|-|
> |Uni-MuMER⁺(w/CJK)|79.46|
> |Uni-MuMER⁺(w/o CJK)|80.45|
>
> **We will include the above details** in Section 4.2 Main Results in the final version.
>
>
> 3. **Clarification of “External Data” Definition**
>
> > “External data” is mentioned in both Table 1 and Table 2. Could the authors clarify which datasets or sources are included in this category?
>
> **Although we mentioned** in Section 4.2 (lines 247), we realize that it was unclear.
>
> As detailed in Appendix B.1 (Paragraph Data Construction, lines 1007–1014), the “external data” collectively includes CROHME, CROHME 2023, HME100K, MathWriting, and Im2Latexv2, totaling 392K images. These datasets, in combination with three tasks (Tree-CoT, EDL, and SC), form the 1.6M samples.
>
> We will clarify this in Section 4.2 to avoid confusion, and open-source our dataset.
>
>
> ## Limitations
>
> **Hallucination and Repetition Issues**
>
> > Do fine-tuned large VLMs like Uni-MuMER suffer from hallucination compared to specialized models? This limitation should be discussed.
>
> Both fine-tuned large VLMs and specialized models are probabilistic model, making them possible to  hallucination and repetition issues.
>
> To quantitatively analyze this, we preliminary adopt two metrics: **Hallucination Token Rate (HTR)** (percentage of predicted tokens not in the ground truth) and **Missing Token Rate (MTR)** (percentage of ground truth tokens not in the pred). Additionally, we categorized hallucinations by token type (numerical(0-9), variable(a-zA-Z).
>
>
> Experimental results (without external data) on the CROHME 2014, 2016, and 2019 datasets clearly demonstrate that Uni-MuMER (HTR=1.18%) **significantly reduces hallucination** compared to both baseline and specialized models.
>
> ||ExpRate|HTR|MTR|HTR_num|HTR_var|
> |-|-|-|-|-|-|
> |CoMER (SOTA'22)|59.18|2.55|3.91|0.37|0.33|
> |TAMER (SOTA'24)|60.29|2.17|4.02|0.29|0.29|
> |Vanilla|68.64|4.08|2.42|1.08|0.29|
> |+Tree-CoT|**70.85**|**1.67**|**1.56**|0.14|0.21|
> |+EDL|70.30|2.58|1.94|1.09|0.22|
> |+CAN|69.86|2.15|2.06|**0.23**|0.27|
> |Uni-MuMER |73.29|1.18|1.39|0.12|0.15|
>
> **We will include the above details** in Section 5 Analysis in the final version.
>
> We sincerely appreciate the reviewer’s constructive suggestions. The additional experiments conducted based on your feedback have strengthened our paper. We hope these improvements provide clear grounds for reconsidering your evaluation and potentially raising your score.

---

> > ### Author Response · Authors · 2025-08-05
> > **Thank You for Your Feedback – Looking Forward to Further Comments**
> >
> > Thank you very much for your detailed review and valuable feedback on our manuscript. We have noted that the discussion deadline is August 6th (AoE). If you have any further comments, questions, or suggestions, please don't hesitate to share them with us. We would be glad to address them promptly and make any necessary improvements. We greatly appreciate your insightful contributions and look forward to refining our paper further based on your valuable advice.
> >
> > Please do not hesitate to reach out if you require any additional information or clarification. We remain committed to actively engaging in the revision process and responding to your feedback thoroughly.

---

### Official Review · Reviewer_uB1H · 2025-07-02

**Clarity:** 4
**Significance:** 3
**Originality:** 3
**Rating:** 5
**Confidence:** 5

**Summary:**

The paper addresses the challenge of Handwritten Mathematical Expression Recognition (HMER) by introducing Uni-MuMER, a framework that fine-tunes the Qwen2.5-VL-3B vision-language model (VLM) with three data-driven tasks: Tree-CoT, Error-Driven Learning (EDL), Symbol Counting (SC). The experiment result shows the Uni-MuMER achieves significant improvement and establishes a new SOTA on the HMER task.

**Questions:**

Have you done some error analysis? Which types of symbols remain most challenging for Uni-MuMER?
Are there more training tasks that can be used to further improve the model？

**Ethical Concerns:**

["NO or VERY MINOR ethics concerns only"]

**Limitations:**

yes. The authors have adequately addressed the limitations.

**Quality:**

4

**Strengths And Weaknesses:**

【strengthes】
1. Unified and efficient framework: Leveraging existing pre-trained VLMs is efficient and convenient to unify multi-tasks to enhance the HMER.
2. Strong performance: the three training tasks work well and largely improve the VLM's HMER capability.
【weaknesses】
For the easy, clear, reasonable, and effective method proposed in the paper, there are not many weaknesses. The dominating weakness is the inference speed and the computational cost increment introduced by the general-purpose large language model when compared with the previous small and specific models.

---

> ### Author Rebuttal · Authors · 2025-07-31
>
> # Response to Reviewer uB1H
>
> We thank the reviewer for the positive assessment of our work. We are glad that the reviewer found our unified and efficient framework, as well as the strong performance improvements of Uni-MuMER. We address each weakness, question, and point raised by the reviewer in detail below.
>
> ## Weakness: Inference Speed and Computational Cost
>
> > For the easy, clear, reasonable, and effective method proposed in the paper, there are not many weaknesses. The dominating weakness is the inference speed and the computational cost increment introduced by the general-purpose large language model when compared with the previous small and specific models.
>
> We appreciate the reviewer’s valid concern about inference speed and computational cost. Indeed, **using a general-purpose VLM incurs higher computational overhead** compared to prior lightweight models.
>
> Our approach, which leverages the optimized **vLLM inference engine**, significantly mitigates this issue. By using some GPU (like A800 80GB), our method achieves **8.29× faster inference speed** compared to previous smaller, specialized models, as mention in Appendix C.2 Inference Efficiency Table 8.
>
> By **retaining a standard VLM architecture without structural modifications**, we fully exploit community-optimized inference framework, which futher highlighting a key strength of our work.
>
> Besides, our empirical results demonstrate that this cost brings significant **performance and speed improvement.**
>
> ## Question1: Error Analysis
>
> > Have you done some error analysis?
>
> Thank you for the suggestion. To deepen our analysis of error types in mathematical expression recognition, we have introduced definitions for hallucination and omission at the token level:
>
> - Hallucination Token Rate (**HTR**): the rate of tokens predicted that do not appear in the ground truth.
>
>   - _HTR_num:_ hallucinated numeric digits (0–9)
>   - _HTR_var:_ hallucinated alphabetic variables (a–z, A–Z)
>   - _HTR_delimiter:_ hallucinated structural delimiters (e.g., [], {}, ())
> - Missing Token Rate (**MTR**): the rate of tokens ground truth that do not appear in the predicted to quantify omitted tokens
>
> We evaluated these metrics across the CROHME 2014, 2016, and 2019 test sets, ensuring that no external data was used during training. The results are summarized below:
>
> | Model                 | ExpRate   | @CDM      | HTR      | MTR      | HTR_num  | HTR_var  | HTR_delim |
> | --------------------- | --------- | --------- | -------- | -------- | -------- | -------- | --------- |
> | CoMER (SOTA 2022)     | 59.18     | 62.47     | 2.55     | 3.91     | 0.37     | 0.33     | 1.18      |
> | TAMER (SOTA 2024)     | 60.29     | 63.18     | 2.17     | 4.02     | 0.29     | 0.29     | 1.05      |
> | Vanilla (baseline)    | 68.64     | 71.74     | 4.08     | 2.42     | 1.08     | 0.29     | 1.68      |
> | Vanilla+Tree-CoT      | **70.85** | 73.71     | **1.67** | **1.56** | 0.14     | **0.21** | 0.85      |
> | Vanilla+EDL           | 70.30     | 73.76     | 2.58     | 1.94     | 1.09     | 0.22     | **0.83**  |
> | Vanilla+CAN           | 69.86     | **74.76** | 2.15     | 2.06     | **0.23** | 0.27     | 1.08      |
> | Uni-MuMER (w/3 tasks) | 73.29     | **77.19** | **1.39** | **1.18** | **0.12** | **0.15** | **0.6**   |
>
>
>
>
> **Uni-MuMER significantly reduces hallucinated tokens and omissions** compared to both the vanilla baseline and prior SOTA specialized models.
>
> - **Tree-CoT** improves structural reasoning and reduces structural hallucinations
> - **EDL** corrects frequent symbol-level mistakes,
> - **SC** enforces token consistency, reducing omissions and hallucinated insertions.
>
> We will incorporate this analysis in Section 5 of the final paper to highlight Uni-MuMER’s robustness.
>
> ## Question2: Which types of Symbols are Most Challenging?
>
> > Which types of symbols remain most challenging for Uni-MuMER?
>
> Thank you for raising this insightful question, as **it pertains directly to inherent challenges within the HMER domain and future work** (response Question 3).
>
> We have conducted a detailed analysis of the error cases from the CROHME datasets. Our observations indicate that the majority of **remaining errors can be attributed primarily to two factors**:
>
> 1. **Hallucination**: As discussed previously (Response to Question 1), **hallucination is a known issue inherent to VLMs**. While **Uni-MuMER substantially alleviates this issue**—as demonstrated by the significantly reduced Hallucination Token Rate (HTR) shown previously—it is challenging to entirely eliminate due to inherent ambiguities in handwritten data.
> 2. **Similar-character (Token-level) Errors**: Another primary error source stems from visually similar or ambiguous characters. Without explicit context, distinguishing certain symbol pairs—such as:
>
>    - Lowercase "s" and uppercase "S" (`s_a` vs. `S_a`),
>    - Digit "0" and uppercase letter "O",
>    - Lowercase letter "a" and Greek letter (`\alpha`),
>    - Digit "1" and lowercase "l",
>      is particularly challenging. These symbol pairs often share substantial visual similarity, making isolated recognition inherently ambiguous even when trained on extensive datasets containing diverse handwriting styles.
>
> Addressing these challenges transcends the traditional exact-match metrics (e.g., ExpRate) and relies on more sophisticated context-aware evaluation methods beyond token-level accuracy.
>
> ## Question3: Additional Training Tasks for Improvement
>
> > Are there more training tasks that can be used to further improve the model？
>
> Looking ahead, we envision that improvements in the model’s integration of broader world knowledge could significantly alleviate issues mentioned in Response Question 2. As the model gains more extensive and nuanced reasoning capabilities, it can better infer plausible characters within ambiguous contexts. Moreover, as mentioned in Section 6 (Discussion), promising directions include:
>
> - **R1-like Reflective Self-Correction:**
>   Allowing the model to iteratively reflect on and correct its predictions, thereby reducing errors due to ambiguous or visually similar symbols.
> - **O3-like Functional Call Visual Reasoning:**
>   Enabling explicit visual reasoning steps, such as zooming into particular regions of an expression image to resolve fine-grained details and mitigate visual ambiguity.
> - **Hierarchical Recognition with Integrated Reasoning:**
>   Structuring the recognition task hierarchically—first counting symbol, then constructing the Abstract Syntax Tree (AST), and embedding reflective and functional-call reasoning (R1-like and O3-like) within the AST construction process to ensure accuracy and structural consistency.
>
> We plan to explore these exciting avenues in future research, extending Uni-MuMER’s capabilities while leveraging the strength and flexibility of our existing framework.

---

> ### Author Response · Authors · 2025-08-05
> **Thank You for Your Feedback – Looking Forward to Further Comments**
>
> Thank you very much for your detailed review and valuable feedback on our manuscript. We have noted that the discussion deadline is August 6th (AoE). If you have any further comments, questions, or suggestions, please don't hesitate to share them with us. We would be glad to address them promptly and make any necessary improvements. We greatly appreciate your insightful contributions and look forward to refining our paper further based on your valuable advice.
>
> Please do not hesitate to reach out if you require any additional information or clarification. We remain committed to actively engaging in the revision process and responding to your feedback thoroughly.

---

### Official Review · Reviewer_QkdQ · 2025-07-02

**Clarity:** 3
**Significance:** 3
**Originality:** 1
**Rating:** 5
**Confidence:** 5

**Summary:**

The paper proposes a multi-task training setup for handwritten math expression recognition which involves basic HMER, syntax tree parsing, symbol counting, and error identification in the OCR output from other models. Authors finetune QWEN2.5VL-3B on these tasks on a range on datasets (CROHME14/16/19, NME, HME100k, MathWriting) and show state-of-the-art results, particularly highlighting CROHME datasets. Authors perform an ablation study to show that the effect of multi-task training helps performance.

**Questions:**

You mention that the performance does rely heavily on having the target dataset in the training mixture, but that this does depend on the variety in the data. One thing that could strengthen the work could be showing the generalization ability, ex. by looking how does the model perform in an odd-one-out setting, where it is trained on N-1 datasets and is evaluated on the Nth - do you have some observations around this?

**Ethical Concerns:**

["NO or VERY MINOR ethics concerns only"]

**Final Justification:**

I thank the authors for their response, specifically the odd-one-out-experiment and the additional results with UNIMerNET and ICAL, which strengthen the paper. I do not believe that the observation that "more data yields better results" is unique or surprising in the HMER case, but given other points addressed by the authors, I am increasing my rating to accept.

**Limitations:**

--

**Paper Formatting Concerns:**

--

**Quality:**

3

**Strengths And Weaknesses:**

Quality & Clarity: The paper is well-written and easy to understand. The appendix is quite comprehensive with additional results, explanations, and code. However, the comparison section lacks comparisons with works like UNIMer and ICAL (Authors do mention the see works in the related work section, and, IIUC, the numbers for those will be lower for CROHME datasets than the ones reported by the authors, but potentially higher than other baselines reported by them, but it would nevertheless strengthen the paper to include them in the table 1). Furthermore, authors only report results on MNE, HME100k, and Mathwriting in the appendix, and on a limited subset of papers - it would be beneficial to see a wider comparison, whenever available, and in the main manuscript.

Originality: The core of the work is a multi-task training setup on a number of datasets, with the observation that multi-task training helps, and having the training dataset corresponding to the test is beneficial for performance (l. 341) - the novelty is quite limited, with the exception of the specific task setup.

Significance: The model does obtain state-of-the-art results and the multi-task training setup is useful for practitioners. This is also complemented by the code in the supplementary material related to data management, training and inference, fostering reproducibility.

---

> ### Author Rebuttal · Authors · 2025-07-31
>
> # Response to Reviewer QkdQ
>
> We thank the reviewer for the thoughtful and constructive feedback. We address each concern and question in detail below, aiming to clarify our design choices and highlight the contributions of our work.
>
> ## Weakness1 **Comparisons with UNIMer and ICAL in Table 1**
>
> > However, the comparison section lacks comparisons with works like UNIMer and ICAL (Authors do mention the see works in the related work section, and, IIUC, the numbers for those will be lower for CROHME datasets than the ones reported by the authors, but potentially higher than other baselines reported by them, but it would nevertheless strengthen the paper to include them in the table 1).
>
> We appreciate the reviewer highlighting these missing comparisons. **Due to page constraints**, we were initially unable to include these results. In the revised manuscript, **we will incorporate comparisons with UnimerNet and ICAL in the updated Table 1** to strengthen our evaluation:
>
> |Model|CROHME 2014 ExpRate|CROHME 2014 ExpRate@CDM|CROHME 2016 ExpRate|CROHME 2016 ExpRate@CDM|CROHME 2019 ExpRate|CROHME 2019 ExpRate@CDM|CROHME Average ExpRate|CROHME Average ExpRate@CDM|
> |--|--|--|--|--|--|--|--|--|
> |ICAL|60.63|-|58.79|-|60.51|-|59.98|-|
> |UnimerNet†|67.4|-|68.4|-|65.4|-|67.07|-|
> |Vanilla|72.62|75.66|64.26|67.83|69.14|71.98|68.67|71.82|
> |Uni-MuMER|75.36|79.11|70.79|75.24|73.73|77.23|73.29|77.19|
> |Uni-MuMER†|**82.05**|**85.09**|**77.94**|**81.08**|**79.23**|**82.40**|**79.74**|**82.86**|
>
>
> ## Weakness 2 **Limited Dataset Evaluation Reporting in the Main Text**
>
> > Furthermore, authors only report results on MNE, HME100k, and Mathwriting in the appendix, and on a limited subset of papers - it would be beneficial to see a wider comparison, whenever available, and in the main manuscript.
>
> We appreciate the reviewer’s suggestion. Due to **strict page limits** and the fact that **some test sets** (e.g., MNE, MathWriting) **are not yet widely tested in prior relative work**, we included these results in the appendix. **We agree** that presenting a broader set of comparisons in the main text will strengthen the paper, and **we will move key results from these datasets into the main paper** in the camera-ready version.
>
> ## Originality: **Novelty is Limited**
>
> > The core of the work is a multi-task training setup on a number of datasets, with the observation that multi-task training helps, and having the training dataset corresponding to the test is beneficial for performance (l. 341) - the novelty is quite limited, with the exception of the specific task setup.
>
> We appreciate the reviewer’s comment regarding novelty.
>
> ### **(1) Novelty from Paradigm Shift**:
>
> From past viewpoint with smaller models, framework innovation was essential; But for the **current generation** of **large-scale VLMs**, we argue that **data-centric and unified multi-task fine-tuning itself constitutes a significant innovation.**
>
> Although each innovation individually offers distinct advantages, as discussed in the paper (line 41), **prior architectural innovations are fragmented and difficult to integrate, which limits the development of HMER domain.**
>
>
> Previous works primarily experimented with smaller models. Their effectiveness when applied to modern large-scale pretrained VLMs has not been validated, and their improvements may not translate to current VLMs
>
> ### **(2) Empirical Novelty (Data Utilization Efficiency)**:
>
> Our method enhances **data utilization efficiency**. Such advantages are broadly applicable to other domains facing challenges with limited or diverse training resources.
>
> ### **(3) Generalization beyond Task-Specific Datasets**:
>
> Moreover, the reviewer **correctly observes that more data generally improves performance**. Yet, we stress that this outcome is **non-trivial and unique** in the context of HMER:
>
> - traditional lightweight models (e.g., CoMER) have **limited benefits or even performance degradation**. As shown explicitly in our ablation (Appendix C.1 **Effects of Data Diversity Compared with Specialized HMER Model**, Table 7), incremental dataset scaling substantially benefits our model (Uni-MuMER), whereas the performance of traditional specialized models is unstable or declines:
>
>     |Model|CROHME Train|+ CROHME-23|+ HME100K|+ MathWriting|+ Im2Latexv2|
>   |-|--|--|--|--|--|
>   |CoMER|58.14|60.08|63.10|52.17|52.29|
>   |Uni-MuMER|**75.36**|**76.27**|**78.01**|**83.20**|**82.86**|
>
> - Additionally, prior studies typically experimented on limited datasets. We found that the performance of these lightweight models often decreases significantly (as demonstrated in Appendix C.1 Table 7).
>
> ## Questions: **Cross-Dataset Generalization – Odd-One-Out Experiment**
>
> > You mention that the performance does rely heavily on having the target dataset in the training mixture, but that this does depend on the variety in the data. One thing that could strengthen the work could be showing the generalization ability, ex. by looking how does the model **perform in an odd-one-out setting**, where it is trained on N-1 datasets and is evaluated on the Nth - do you have some observations around this?
>
> We thank the reviewer for this insightful suggestion. As suggested, **we conducted an “odd-one-out” generalization experiment**, training Uni-MuMER on N-1 datasets and evaluating it on the held-out N-th dataset.
>
> The results below present the expression recognition accuracy (ExpRate@CDM%) in each scenario. For comparison, we include the baseline CoMER trained on all datasets:
>
>
> | Model                               | CROHME-Avg | CROHME 2023 | HME100K | MathWriting | Im2LaTeXv2 |
> |-------------------------------------|------------|-------------|---------|-------------|------------|
> | Doubao-1.5-pro                      | 65.77      | 58.18       | 55.08   | 26.34       | 27.66      |
> | CoMER⁺ (w/o CROHME)                 | 45.99      | 58.28       | 39.49   | 23.23       |            |
> | CoMER⁺                              | 52.29      | 59.91       | 44.63   | 28.45       |            |
> | Uni-MuMER⁺ *(w/o CROHME)*           | *78.22*      | 72.19       | 74.47   | 68.08       | 89.18      |
> | Uni-MuMER⁺ *(w/o CROHME-2023)*      | 82.31      | *71.66*      | 74.68   | 68.85       | 89.77      |
> | Uni-MuMER⁺ *(w/o HME100K)*          | 82.30      | 72.01       | *52.38*   | 68.94       | 91.19      |
> | Uni-MuMER⁺ *(w/o MathWriting)*      | 79.58      | 73.51       | **74.29** | *32.55*       | **93.31**  |
> | Uni-MuMER⁺ *(w/o Im2LaTeXv2)*       | **83.20**  | 73.33       | 72.40   | 68.93       | *70.06*      |
> | Uni-MuMER⁺                          | 82.89      | **74.42**   | 72.31   | **69.11**   | 88.99      |
>
>
> **Observations:**
>
> Our results clearly demonstrate the strong generalization capability of Uni-MuMER compared to the baseline model (CoMER):
>
> - **Even without training on the target domain**, Uni-MuMER consistently achieves substantially higher accuracy than the baseline CoMER† trained on **all datasets**. For example, Uni-MuMER without MathWriting achieves 32.55% accuracy on MathWriting, significantly surpassing CoMER’s 26.34% accuracy (trained on MathWriting explicitly).
>
> This finding confirms that Uni-MuMER **effectively leverages shared knowledge across domains and generalizes well to unseen datasets**, validating its strong multi-task learning capabilities.
>
> However, we also notice that the degree of generalization depends on data distribution similarity:
>
> - For datasets closely related to others (e.g., CROHME and CROHME2023), accuracy drops are minimal (generally within 1-3%), further indicating good cross-domain transfer.
> - For datasets with distinct characteristics (e.g., HME100K for **real-life, low-quality**, MathWriting for **densely structured expression**, and Im2LaTeXv2 for **printed expression**), accuracy drops are more pronounced. Nevertheless, even in these challenging cases, Uni-MuMER outperforms baseline methods significantly.
>
> In summary, our “odd-one-out” experiments demonstrate that Uni-MuMER exhibits remarkable generalization capability and robustness compared to previous methods, **validating the benefits of the proposed multi-domain modeling approach**.
>
> We will clearly highlight these findings in the revised manuscript and further discuss the implications and limitations of our approach.

---

> > ### Author Response · Authors · 2025-08-05
> > **Thank You for Your Feedback – Looking Forward to Further Comments**
> >
> > Thank you very much for your detailed review and valuable feedback on our manuscript. We have noted that the discussion deadline is August 6th (AoE). If you have any further comments, questions, or suggestions, please don't hesitate to share them with us. We would be glad to address them promptly and make any necessary improvements. We greatly appreciate your insightful contributions and look forward to refining our paper further based on your valuable advice.
> >
> > Please do not hesitate to reach out if you require any additional information or clarification. We remain committed to actively engaging in the revision process and responding to your feedback thoroughly.

---

### Official Review · Reviewer_DUPV · 2025-07-02

**Clarity:** 4
**Significance:** 3
**Originality:** 3
**Rating:** 5
**Confidence:** 4

**Summary:**

This paper explores the use of a general-purpose large language model (vLLM), specifically Qwen, fine-tuned for the task of handwritten mathematical expression recognition. The authors introduce three targeted fine-tuning tasks designed to inject task-specific knowledge into the model: (1) generation of abstract syntax trees (ASTs), (2) self-error detection and correction, and (3) symbol counting.

The proposed approach is evaluated against both state-of-the-art specialized models and larger generalist models on several databases. Experimental results show that the fine-tuned Qwen model outperforms these baselines across several benchmarks. The paper also presents an ablation study assessing the individual contribution of each fine-tuning task as well as an analysis of the model's out-of-domain generalization capabilities.

**Questions:**

- Why choosing Qwen-3B for fine-tuning, considering that the 7B version already achieves significantly better performance even without fine-tuning?
- In Table 8, inference time comparisons suggest that smaller specialized models are slower than Qwen. How is this possible?
- Will the source code and trained models be made publicly available?
- What value of k was used in the k-fold Error Corpus Generation process?
- In Table 3, the dataset version is not specified. Moreover, I could not locate the reported result for CoT+EDL+SC (73.29) in other tables, particularly Table 1. Could you clarify this point?
- The experiments in Section 5.2 suggest that the specialization of the Qwen model is only effective when fine-tuning is performed with in-domain data. Simple task-specific fine-tuning (for mathematical expression recognition) appears insufficient unless the fine-tuning data is closely aligned with the test set. This observation seems to partially contradict the goal of the proposed specialization tasks, which aim to introduce general task knowledge into the model, independent of the dataset. Could the authors clarify their position on this point and discuss whether in-domain fine-tuning remains, in practice, the most effective way to improve model performance?
- The proposed future direction of Self-Correction for Expression Recognition remains somewhat unclear. It is difficult to see how "colloquial expressions," such as the example provided, could be effectively integrated, given that they depend on the specific expression being recognized. Could the authors provide more concrete examples or clarify how this type of contextual, expression-dependent correction could be implemented?

**Ethical Concerns:**

["NO or VERY MINOR ethics concerns only"]

**Limitations:**

The authors have addressed some of the limitations of their approach, notably by discussing how their model could be evaluated on other tasks, such as OCR. However, the paper does not adequately address the potential implications of using a significantly larger generalist language model for a relatively narrow, well-defined task such as handwritten mathematical expression recognition. The increased computational resources and environmental costs associated with large-scale vLLMs could limit the practicality of the approach compared to smaller, task-specific models. A more explicit discussion of these issues would improve the paper.

**Quality:**

4

**Strengths And Weaknesses:**

## Strengths:
- First systematic evaluation of fine-tuning a generalist vLLM for handwritten mathematical expression recognition.
- Well-motivated fine-tuning tasks (AST generation, self-correction, symbol counting) that are relevant and can be adapted to other domains.
- The model outperforms both specialized state-of-the-art models and larger generalist models.
- Ablation study (Section 5) is carefully designed and informative, clearly linking fine-tuning tasks to specific recognition improvements.

## Weaknesses:
- The scientific novelty of using a vLLM for this task is limited, even though the experimental exploration is solid.
- The effectiveness of the approach seems heavily dependent on access to in-domain data, which limits its general applicability.
- The Error-Driven Learning approach requires fine-tuning multiple models, making it potentially expensive to apply.
- The future research direction on self-correction lacks clarity, especially regarding handling of context-dependent expressions.

---

> ### Author Rebuttal · Authors · 2025-07-31
>
> # Response to Reiewer DUPV
>
> We sincerely thank the reviewer for the thoughtful feedback, which is helpful to improve the paper.
>
> ## **Weakness 1:** **Limited Novelty of Using a vLLM**
>
> > The scientific novelty of using a vLLM for this task is limited, even though the experimental exploration is solid.
>
> Due to space limitations, please refer to Reviewer QkdQ’s comments regarding Originality for further discussion on this issue.
>
>
> ## **Weakness 2: Dependence on In-Domain Data**
>
> > The effectiveness of the approach seems heavily dependent on access to in-domain data, which limits its general applicability.
>
> Thank you for highlighting this point. To investigate our model’s generalization, we conducted an additional “Odd-One-Out” experiment (training on N-1 datasets, testing on the held-out dataset), evaluted with ExpRate@CDMmailto:ExpRate@CDM %.
>
>
> |Model|CROHME-Avg|CROHME 2023|HME100K|MathWriting|Im2LaTeXv2|
> |-|-|-|-|-|-|
> |CoMER⁺(w/oCROHME)|45.99|58.28|39.49|23.23||
> |CoMER⁺|52.29|59.91|44.63|28.45||
> |Uni-MuMER⁺*(w/oCROHME)*|*78.22*|72.19|74.47|68.08|89.18|
> |Uni-MuMER⁺*(w/oC-2023)*|82.31|*71.66*|74.68|68.85|89.77|
> |Uni-MuMER⁺*(w/oHME100K)*|82.30|72.01|*52.38*|68.94|91.19|
> |Uni-MuMER⁺*(w/oMathWrt)*|79.58|73.51|**74.29**|*32.55*|**93.31**|
> |Uni-MuMER⁺*(w/oIm2LTX)*|**83.20**|73.33|72.40|68.93|*70.06*|
> |Uni-MuMER⁺|82.89|**74.42**|72.31|**69.11**|88.99|
>
> The odd-one-out evaluation confirms that our unified model can **generalize to an unseen dataset reasonably well**. When the unseen domain is very different, performance drops more markedly, which highlights an area for improvement. HME100K for **real-life, low-quality** images, MathWriting for **densely structured expression**, and Im2LaTeXv2 for **printed expression images**, each of which represents a markedly different and challenging domain.
>
> Crucially, after introducing even modest amounts of domain-relevant data, **our unified approach easily achieves top-tier performance.** Thus, our results indicate that, compared to existing specialized models and general VLM baselines, our approach provides superior generalization with **minimal domain-specific data requirements**, highlighting a clear advantage and direction for future improvement.
>
> We will include these findings in the paper, discuss the implications for generalization and acknowledge the limitations.
>
> ## **Weakness 3 : Computational Cost of Error-Driven Learning (EDL) with Multiple Models**
>
> Thank you for raising this important point. To balance effectiveness and computational efficiency, we chose a modest value (k=3) for the k-fold setup, considering both data quality and training costs. While this does increase computational cost, we believe it is justified because:
>
> 1. **Error-Driven Learning** provides a general, reusable strategy enabling continuous model improvement.
> 2. The resulting **error corpus**, which will be open-sourced, contains inherently valuable, challenging examples tailored specifically to our strongest current model (Uni-MuMER), offering lasting benefits for future HMER research.
>
> ## **Weekness 4 : Clarity of Future Work on Self-Correction**
>
> > The future research direction on self-correction lacks clarity, especially regarding handling of context-dependent expressions.
>
> Thank you for your interest in our proposed future direction on self-correction.
>
> Uni-MuMER introduces a novel perspective on enabling **self-reflective behavior** for mathematical expression recognition:
>
> 1. Uni-MuMER brings a **Tree-aware Chain-of-Thought** mechanism and basic error-correction capabilities. Combining these aspects can naturally facilitate self-reflection behaviors, similar in spirit to **Deepseek-R1**.
>
> 2. Specifically, we can obtain data with error correction behavior through synthesis or manual annotation to train the model to acquire similar capabilities, and then enhance it through RL. (Similar to the process of Deepseek-R1)
>
> 3. To illustrate how our envisioned method handles context-dependent expressions, consider the example expression `\lim _ { n \to \infty } ( \sum _ { k = 1 } ^ n \frac 1 { k ^ 2 } )`: $lim _ { n \to \infty } ( \sum _ { k = 1 } ^ n \frac { 1 } { k ^ 2 } )$
>
> - Without Tree-CoT and error-correction, the model might mistakenly decode the final token as “2”.
>
> - With Tree-CoT and self-correction capabilities, upon initially decoding the token as “2”, the model can reflect and identify an unclosed left parenthesis earlier in the expression. By systematically reviewing the tree structure (as enabled by Tree-CoT), it can correct this token, determining that a right parenthesis is contextually more appropriate than “2”.
>
> This example highlights the practical benefit of our envisioned approach for handling context-dependent expressions, and we will include a clearer description of this idea in the revised manuscript.
>
>
>
> ## Question 1 **Base Model Selection (Why Qwen-2.5-VL-3B?)**
>
> > Why choosing Qwen-3B for fine-tuning, considering that the 7B version already achieves better than Uni-MuMER?
>
> Thank you for raising this question. We chose **Qwen2.5-VL-3B** to strike a balance between **performance and computational cost**, ensuring our approach remains suitable for real-world deployment.
>
> Importantly, we also **trained a 7B version** of Uni-MuMER for comparison and observed only **marginal improvements:**
>
>
> |Model|CROHME-Average|CROHME 2023 test|HME100K|Mathwriting test|Im2Latexv2 test|
> |--|--|--|--|--|--|
> |Uni-MuMER†|80.45|**46.35**|72.72|51.41|88.03|
> |Uni-MuMER-7B†|**80.55**|46.04|**73.25**|**53.33**|**91.32**|
>
>
>  Additionally, as noted in Table 1, **even the larger zero-shot models** (7B, 72B, Gemini2.5-flash) significantly **underperform compared to specialized HMER methods**, while our fine-tuned 3B model surpasses them by **over 20%**. This strongly indicates that effective domain-specific fine-tuning matters more for HMER than the base model size, further justifying our choice of the smaller, more efficient **3B** version.
>
>
> ## Question 2 **Inference Time in Table 8**
>
> > In Table 8, inference time comparisons suggest that smaller models are slower than Qwen. How is this possible?
>
> The higher inference speed of Qwen-based Uni-MuMER is due to the **vLLM inference engine**, which provides **highly optimized batched decoding and caching mechanisms**, including optimized caching mechanisms such as PagedAttention for efficient KV caching. With access to large-memory GPUs (e.g., A800 80GB), vLLM can cache more key-value pairs, further accelerating inference. In contrast, smaller models are often compute-bound rather than memory-bound.
>
> Additionally, previous specialized models typically rely on **beam search**, which is inherently slower than the **nearly greedy decoding** used in VLMs. vLLM performs an **optimized prefill stage**, amortizing the initial encoding cost across multiple subsequent decoding steps and boosting per-sample inference speed in batched processing.
>
> Many specialized models employ **custom architectural modifications** (e.g., tree decoders), which **prevent them from leveraging these general-purpose inference optimizations** available in developer groups.
>
> Our approach **retains standard VLMs architecture** without structural changes **highlighting its key strength.**
>
> ## Question 3 **Open-Sourcing**
>
> > Will the source code and trained models be made publicly available?
>
> Yes, we will make the code, trained models, training data and reproducible training scripts publicly available.
>
>
>
> ## Question 4 **k-Fold Value in Error-Driven Learning**
>
>
> Thank you for pointing out this oversight. We employed a **3-fold** on the training set to generate corpus, which was **chosen as a balance** **between generating** error corpus and **keeping computation** feasible.
>
> We will clarify this detail in Section 3.3 in the final version.
>
>
>
> ## Question 5 **Clarification on Table 3 (Ablation Results)**
>
> > In Table 3, the dataset version is not specified. Moreover, I could not locate the reported result for CoT+EDL+SC (73.29) in other tables, particularly Table 1. Could you clarify this point?
>
>
>
> The result of 73.29% ExpRate corresponds to the Uni-MuMER (w/o †) row in Table 1, under the “Average Exp” column (**second-to-last row, second-to-last column**).
>
> This version of **Uni-MuMER** includes all three proposed tasks (Tree-CoT, EDL, and SC) and **is aligned with the “Vanilla+CoT+EDL+SC” setting in Table 3**. To **ensure a fair comparison with previous work**, this model was fine-tuned only on the CROHME training set, without any external data.
>
>
>
> ## Question 6 **Generalization vs. In-Domain Performance**
>
> > The experiments in Section 5.2 suggest that the specialization of the Qwen model is only effective when fine-tuning is performed with in-domain data...
>
>
>
> Thank you for your comment. We appreciate the opportunity to clarify.
>
> Our intended message was that fine-tuning effectiveness indeed varies based on the closeness of the training and test datasets:
>
> - **In-domain fine-tuning consistently achieves the highest performance**. For example, printed-expression data fine-tuning significantly outperforms using handwritten data alone (90% vs. 70% ExpRate@CDM).
>
> - **Partial out-of-domain data can still improve performance**, as illustrated in Figure 6, where printed data enhanced some handwritten datasets.
>
> - **Mismatched datasets may degrade performance** due to substantial domain gaps, such as training on clean expressions and testing on low-quality real-life images (e.g., HME100K), may degrade performance due to substantial domain gaps.
>
> Additionally, we are actively **exploring the integration of general SFT datasets** (e.g., LLaVA’s One Vision data) **with our HMER-specific training data to assess broader generalization performance.** Due to computational constraints, this evaluation is ongoing, and we plan to discuss these findings further in future revisions.
>
> ## Limitaions
>
> Due to space limitations, detailed explanations of future work and limitations will be provided later.

---

> > ### Author Response · Authors · 2025-08-01
> >
> > ### Question 7 **“Self-Correction” Mechanism in Future Work**
> >
> > > The proposed future direction of Self-Correction for Expression Recognition remains somewhat unclear. It is difficult to see how "colloquial expressions," such as the example provided, could be effectively integrated, given that they depend on the specific expression being recognized. Could the authors provide more concrete examples or clarify how this type of contextual, expression-dependent correction could be implemented?
> >
> >
> >
> > Thank you for your insightful comment.
> >
> > Here we present one specific technical approach that can be implemented based on Uni-MuMER. We outline a concrete technical path below:
> >
> > 1. **Multiple Sampling with Tree-CoT**: The model generates multiple output, from the same input, which contains reasoning Tree-aware CoT rather than providing direct recognition results immediately. For example:
> >    - <input\_image\_text>: <tree\_cot\_1><final\_answer\_1> | correct
> >    - <input\_image\_text>: <tree\_cot\_2><final\_answer\_2> | incorrect
> >    - … and so on.
> >
> > 2. **Error Detection**: By comparing generated answers, we **identify incorrect reasoning instances** (e.g., <tree\_cot\_2><final\_answer\_2>).
> >
> > 3. **Constructing Self-Correction Data**: We build corrective datasets by explicitly prompting the model with reflective context. For instance:
> >    - Concatenate incorrect reasoning with corrective feedback:
> >    - <tree\_cot\_2><final\_answer\_2> + "Wait, this seems incorrect because..." + <analysis> + "Therefore, the correct answer should be " + <golden\_tree\_cot><golden\_tree\_answer>.
> >
> > 4. **Training for Reflective Capability**: By training the model on this self-correction dataset, we cultivate an inherent self-reflective behavior. Reinforcement learning further strengthens and generalizes this behavior.
> >
> >
> >
> > This self-reflective behavior is the foundational capability proposed by Uni-MuMER's thought chain and error correction mechanism.
> >
> > Some supplementary descriptions: When humans identify formulas—especially handwritten formulas—they may also be unable to directly and accurately recognize certain characters. However, we can sometimes infer what is more likely to be present in that position.
> >
> > ## Limitations
> >
> > **Scalability and Resource Usage**
> >
> > > The authors have addressed some of the limitations of their approach, notably by discussing how their model could be evaluated on other tasks, such as OCR. However, the paper does not adequately address the potential implications of using a significantly larger generalist language model for a relatively narrow, well-defined task such as handwritten mathematical expression recognition. The increased computational resources and environmental costs associated with large-scale vLLMs could limit the practicality of the approach compared to smaller, task-specific models. A more explicit discussion of these issues would improve the paper.
> >
> >
> >
> > We sincerely appreciate your valuable suggestion regarding scalability and resource usage.
> >
> > We acknowledge the concerns raised about employing a larger generalist model (vLLM) for the HMER task. Below, we concisely clarify how we address these points:
> >
> > 1. **Generalization vs. Efficiency:**
> >    - As detailed in our previous response (see Weakness 1 & Question 6 responses), our approach explores broader task generalization, including preliminary experiments on OCR and integration with general SFT datasets (e.g., LLaVA’s One Vision data).
> >
> > 2. **Model Selection and Computational Cost:**
> >    - Referencing our previous justification (Question 1 response), we explicitly selected the Qwen-2.5-VL-3B model to balance computational efficiency and performance effectively.
> >
> > 3. **Inference Optimization:**
> >    - As discussed in Question 2's response, optimized inference engines like vLLM significantly enhance computational efficiency during inference, balancing initial training costs.
> >
> > 4. **Future Work:**
> >    - We have significantly clarified our future work directions, as discussed in Weakness 4 & Question 7 , explicitly indicating how our approach can be effectively extended to various contextual and colloquial expressions.
> >
> > We will explicitly address these computational trade-offs, resource implications, and practical guidelines in the revised manuscript.
> >
> >
> >
> > Thank you again for your thorough discussion and numerous valuable suggestions for improving our manuscript.

---

> > ### Author Response · Authors · 2025-08-05
> > **Thank You for Your Feedback – Looking Forward to Further Comments**
> >
> > Thank you very much for your detailed review and valuable feedback on our manuscript. We have noted that the discussion deadline is August 6th (AoE). If you have any further comments, questions, or suggestions, please don't hesitate to share them with us. We would be glad to address them promptly and make any necessary improvements. We greatly appreciate your insightful contributions and look forward to refining our paper further based on your valuable advice.
> >
> > Please do not hesitate to reach out if you require any additional information or clarification. We remain committed to actively engaging in the revision process and responding to your feedback thoroughly.

---

> > > ### Comment · Reviewer_DUPV · 2025-08-05
> > >
> > > The authors responded accurately and comprehensively to the questions and comments.
> > >
> > > More and more generalist vLLMs are being fine-tuned for various document analysis tasks (layout, OCR, math, reading order). The question is whether hyperspecialised fine-tuning (e.g. on mathematical expressions) is the best strategy or whether fine-tuning on multiple tasks is better. If multiple tasks are a better strategy, how should the tasks be chosen? This article provides some preliminary answers to these questions.

---

> > > > ### Author Response · Authors · 2025-08-07
> > > >
> > > > Thank you for the important questions.
> > > >
> > > > From the perspective of general‑purpose LLMs and VLMs, mixing data from different domains can yield substantial performance gains [1, 2]. Moreover, when the task mix ratio is chosen carefully, multi-task fine-tuning leads to superior overall performance compared to single-task fine-tuning [3]
> > > >
> > > > Building on the reviewer's question of how to balance multi-task and hyperspecialised fine-tuning, we realized the critical for quantifying the contribution of our proposed tasks:
> > > >
> > > > 1. **Whether multi-task fine-tuning with a combination of general-domain tasks and HMER-specific data is better, and what is the optimal mixture ratio between them?**
> > > > 2. **How do our three proposed tasks impact the model's general capabilities, especially when compared to training on vanilla HMER data?**
> > > >
> > > > To begin addressing the first question, we ran an experiment that compares:
> > > >
> > > > - **Uni‑MuMER:** fine‑tuned only on HMER data;
> > > > - **Uni‑MuMER‑LLAVA:** fine‑tuned on a 1 : 1 mixture of HMER and LLaVA‑OneVision data.
> > > >
> > > > | **Model**              | **MMMU_val** | **Math-Vision** | **MathVista_testmini** |
> > > > | ---------------------- | ------------ | --------------- | ---------------------- |
> > > > | Uni-MuMER              | 47.89        | 24.01           | 47.8                   |
> > > > | Uni-MuMER-LLAVA        | 48.67        | **24.34**       | **51.1**               |
> > > > | Qwen2.5-VL-3B-Instruct | **52.78**    | 21.38           | 47.9                   |
> > > >
> > > > | **Model**       | **CROHME-Avg** | **CROHME-2023** | **HME100K** | **Mathwriting** | **Im2LaTeXv2** |
> > > > | --------------- | -------------- | --------------- | ----------- | --------------- | -------------- |
> > > > | Uni-MuMER-3B    | **82.89**      | **74.42**       | 72.31       | 69.11           | 88.99          |
> > > > | Uni-MuMER-llava | 81.99          | 69.10           | **73.48**   | **70.48**       | **93.44**      |
> > > >
> > > >
> > > >
> > > >
> > > > The results show that **Uni‑MuMER**, despite being fine‑tuned exclusively on HMER, **retains strong general abilities**, with only a slight drop on MMMU. **Uni-MuMER-LLAVA** **improves its overall capabilities** while sacrificing only marginal HMER performance.
> > > >
> > > > While full-scale exploration is constrained by time and resources, we are actively investigating these questions. Our current experiments include:
> > > >
> > > > - How different ratios of HMER to general‑domain data (e.g., 1 : 2, 1 : 5, 1 : 10) influence overall performance;
> > > > - With the amount of general‑domain data fixed, ablation studies of each individual task and their combinations to quantify their respective contributions.
> > > >
> > > >
> > > >
> > > >
> > > >
> > > > [1] Yan, D., Li, Y., Chen, Q. G., Luo, W., Wang, P., Zhang, H., & Shen, C. (2025). Mmcr: Advancing visual language model in multimodal multi-turn contextual reasoning. *arXiv preprint arXiv:2503.18533*.
> > > >
> > > > [2] Ye, J., Liu, P., Sun, T., Zhan, J., Zhou, Y., & Qiu, X. (2024). Data mixing laws: Optimizing data mixtures by predicting language modeling performance. *arXiv preprint arXiv:2403.16952*.
> > > >
> > > > [3] Renduchintala, H. K., Bhatia, S., & Ramakrishnan, G. (2024). SMART: Submodular data mixture strategy for instruction tuning. *arXiv preprint arXiv:2403.08370*.

---

### Author Response · Authors · 2025-08-09
**General Response to All Reviewers (1)**

We thank all the reviewers for their valuable and constructive comments. We have revised the paper as suggested by the reviewers, and have addressed their comments.

Currently, we are particularly grateful to the two reviewers who have actively engaged in discussion and provided insightful feedback. We are still awaiting comments from the other two reviewers.

Below, we clarify several key issues in detail: Paradigm Shift Novelty, generality of Uni-MumER, and concerns about cost.

## **Novelty: Paradigm Shift in HMER**

We wish to reemphasize our focus on the perspective of VLMs rather than small task-specific models. Our work **addresses previous limitations** by introducing **unified solution**, achieving **super SOTA** performance, and establishing a **solid foundation for future** research under this new paradigm.

1. **Comparison with Previous lightweight Models**

We acknowledge that Uni-MuMER does not emphasize architectural innovation, compared to past lightweight models such as TAMER, ABM, PosFormer, and CAN. Those HMER-specific models focus on hard‑code domain priors like tree structures or mitigating inherent model limitations. These tailored designs suffer from notable limitations:

- **Do not compose well**. Different architectural improvements are hard to integrate, limiting overall capability.
- **Gains do not accumulate across datasets**, leading to significant performance degradation when training datasets cover multiple HMER domains.
  Appendix C.1 Table 7: Impact of incrementally training data scaling on CROHME, **scaling data hurts specialized models but helps Uni‑MuMER**

| **Model** | **CROHME Train** | **+ CROHME-23** | **+ HME100K** | **+ MathWriting** | **+ Im2Latexv2** |
| --------- | ---------------- | --------------- | ------------- | ----------------- | ---------------- |
| CoMER     | 58.14            | 60.08           | 63.10         | 52.17             | 52.29            |
| Uni-MuMER | **75.36**        | **76.27**       | **78.01**     | **83.20**         | **82.86**        |

2. **Advantages and Innovations of Uni-MuMER**

By leveraging the strengths of modern VLMs, Uni-MuMER introduces a **unified modeling paradigm** capable of jointly addressing multiple HMER-related tasks, achieving super SOTA performance. Specifically,

- **Integrates multiple traditional HMER tasks within a unified framework**, thoroughly exploring their optimal performances.
- **Introduces error-detection and correction mechanisms**, enabling the model to explicitly discern and learn from subtle correct/incorrect contrasts.
- **Combines strong HMER performance with maintained general capabilities**, as confirmed by additional evaluation of math and multimodal-reasoning benchmarks.

In addition, although not our main highlight, Uni‑MuMER benefits from **general domain pre‑training beyond HMER‑only training**, and achieves **faster inference speed** in practice compared to previous models due to optimized inference frameworks such as vLLM.

3. **Future Directions Enabled by Uni-MuMER**

Looking ahead, Uni-MuMER **lays the groundwork for advanced reflection mechanisms** analogous to DeepSeek-R1, opening pathways for further improvements in HMER. Integrating chain-of-thought reasoning with error detection and correction can **further raise the capability ceiling**. Indeed, we **have outlined** a comprehensive and **inplementable** **scheme** in our response to Reviewer DUPV.

Overall, despite rapid advancements in general pretrained VLMs, HMER **still remains challenging** (current VLMs exhibit poor zero-shot HMER capabilities). Our paper introduces **a new paradigm which replacing fragile HMER architectures with a unified model yields higher accuracy, stronger scalability, and faster inference**. We envision this paradigm as a viable and promising **feasible new direction** for the development of HMER and **contributes to the community.**

---

> ### Author Response · Authors · 2025-08-09
> **General Response to All Reviewers (2)**
>
> ## **Generalization vs in‑domain data**
>
> We discuss our method's generalization capabilities across datasets and domains.
>
> Additional experiments showing that:
>
> - **Uni-MuMER effectively leverages shared knowledge** across domains and **generalizes well** to unseen datasets
> - **Uni‑MuMER retains strong general capabilities. Uni‑MuMER-LLAVA benefits from general domain pre‑training.**
>
> **Odd-one-out experiment**: We **conducted an experiment** which trained on N-1 datasets and evaluated on N (suggested by QkdQ). The results show that Uni-MuMER **effectively leverages shared knowledge** across domains and **generalizes well** to unseen datasets.
>
> |Model|CROHME-Avg|CROHME 2023|HME100K|MathWriting|Im2LaTeXv2|
> |-|-|-|-|-|-|
> |CoMER†(w/oCROHME)|45.99|58.28|39.49|23.23|40.39|
> |CoMER†|52.29|59.91|44.63|28.45|53.37|
> |Uni-MuMER†(w/oCROHME)|*78.22*|72.19|74.47|68.08|89.18|
> |Uni-MuMER†(w/oCROHME-2023)|82.31|*71.66*|**74.68**|68.85|89.77|
> |Uni-MuMER†(w/oHME100K)|82.30|72.01|*52.38*|68.94|91.19|
> |Uni-MuMER†(w/oMathWriting)|79.58|73.51|74.29|*32.55*|**93.31**|
> |Uni-MuMER†(w/oIm2LaTeXv2)|**83.20**|73.33|72.40|68.93|*70.06*|
> |Uni-MuMER†|82.89|**74.42**|72.31|**69.11**|88.99|
>
> **Geneal domain Comparison**: We ran an experiment to validate whether Uni‑MuMER benefits from **general domain data beyond HMER‑only training** (inspired by DUPV). Specifically, we compares:
>
> - **Uni‑MuMER:** fine‑tuned only on HMER data;
> - **Uni‑MuMER‑LLAVA:** fine‑tuned on a 1 : 1 mixture of HMER and LLaVA‑OneVision data.
>
> The results show that **Uni‑MuMER, despite being fine‑tuned exclusively on HMER, retains strong general capabilities.**
>
> For **Uni-MuMER-LLAVA, it improves its overall capabilities** while sacrificing only marginal HMER performance.
>
> |Model|MMMU_val|Math-Vision|MathVista_testmini|
> |-|-|-|-|
> |Uni-MuMER|47.89|24.01|47.8|
> |Uni-MuMER-LLAVA|48.67|**24.34**|**51.1**|
> |Qwen2.5-VL-3B-Instruct|**52.78**|21.38|47.9|
>
> |Model|CROHME-Avg|CROHME-2023|HME100K|Mathwriting|Im2LaTeXv2|
> |-|-|-|-|-|-|
> |Uni-MuMER-3B|**82.89**|**74.42**|72.31|69.11|88.99|
> |Uni-MuMER-llava|81.99|69.10|**73.48**|**70.48**|**93.44**|
>
>
> ---
>
> ## **Efficiency and Cost**
>
> Regarding concerns about increased computational resources, we clarify below the primary resource-consuming stages and the efficiency of our approach in training, error-driven learning (EDL) corpus collection, and inference:
>
> - **Training**: Our model can be effectively trained on 8 A100 GPUs within approximately 20 hours, ensuring manageable training cost.
> - **EDL**: The EDL corpus collection requires computational resources similar to those for the original dataset.  We use k=3-fold cross-validation to balance quality and cost. The open-sourced challenging error/correct pairs provide valuable, reusable resources.
> - **Inference**: Under identical hardware, Uni-MuMER achieves significantly higher throughput compared to traditional specialized models (see Appendix C.2, Table 8). Without architectural modifications, our approach benefits directly from existing community optimizations such as vLLM.
>
> ## **New Experiments**
>
> We have updated the manuscript according to reviewers’ comments. Specifically:
>
> - To clarify the generalization ability of our proposed approach, we conducted an “odd-one-out” experiment on training datasets and compared Uni-MuMER and Uni-MuMER-LLAVA using general multimodal reasoning benchmarks. The results clearly demonstrate **Uni-MuMER’s strong generalization capability.**
> - Regarding concerns about comparability with prior work, we evaluated the impact of filtering and found **minimal effects on comparability.**
> - To address the hallucination issue in fine-tuned large VLMs, we introduced two metrics—Hallucination-Token-Rate (HTR) and Missing-Token-Rate (MTR)—and compared Uni-MuMER with specialized models. Results highlight that **Uni-MuMER significantly reduces hallucinations.**
> - We conducted ablation studies on alternative explorations such as Tree-CoT and Symbol Counting formats. Evaluation results confirm that **our chosen formats are optimal.**
>
> ## **Writing**
>
> We have edited our manuscript to improve the clarity of experiments, figures, and tables. Specifically:
>
> - Explicit discussions on external data have been moved from Appendix B.1 to Section 4.
> - Explicit discussions on Uni-MuMER’s generalization are now included in Sections 5 and 6.
> - Explicit discussions on future directions have been added to Section 6.
> - Key results from additional datasets have been moved into Section 4.
> - Alternative explorations are explicitly discussed in Appendix C.3.
> - Error analysis and discussions on the hallucination issue of VLMs are detailed in Appendices C.4 and E.

---

### Note · Authors · 2025-08-16

We thank the AC and reviewers for their thoughtful feedback and for the constructive discussion during rebuttal. In this summary, we briefly note feedback from reviewers, address key concerns and present concrete updates.

### **Recognized Strengths**

We appreciate that reviewers acknowledged the contributions of our work as "**first systematic evaluation of vLLM for HMER and experimental exploration is solid**" (from Reviewer DUPV),  addressing "**a long-standing challenge in OCR**" (from Reviewer Yrc6), and noted that "**well-written and easy to understand**" (from Reviewer QkdQ),  "**strong performance; easy, clear, reasonable, and effective method**" (from Reviewer uB1H), and "**Well-motivated fine-tuning tasks**" (from Reviewer DUPV, Yrc6).

### **Key Updates**

We have addressed each of the reviewers' comments and identified weaknesses point by point, and we'd also like to highlight a few general improvements we've made to our work based on your comments:

- Based on suggestions by Reviewer QkdQ, we have conducted an “**odd-one-out” experiment** on five training datasets. The result shows that **Uni-MuMER demonstrates strong generalization capability.**
- Inspired by the discussion with Reviewer DUPV , we ran an experiment that compares Uni-MuMER and Uni-MuMER-LLAVA (which mixes data from general domains). The result shows that Uni‑MuMER **benefits from general domain data beyond HMER‑only training**, and **retains strong general capabilities.**
- Based on suggestions by Reviewer Yrc6 and uB1H, we have conducted a quantitative hallucination analysis. The result shows that **Uni-MuMER** significantly **reduces hallucinations** compared to specialized models.

---

### Decision · Program_Chairs · 2025-09-17

**Decision:**

Accept (spotlight)

**Comment:**

(a) This paper introduces Uni-MuMER, a framework that fine-tunes a general-purpose vision-language model (VLM), Qwen-3B, for the specialized task of Handwritten Mathematical Expression Recognition (HMER). The central claim is that a VLM's capabilities for this task can be significantly enhanced through a unified, multi-task fine-tuning approach. The authors propose three novel auxiliary tasks designed to inject domain-specific knowledge: Tree-Aware Chain-of-Thought (Tree-CoT) to improve structural reasoning, Error-Driven Learning (EDL) to enable self-correction capabilities by learning from an error corpus, and Symbol Counting (SC) to enforce recognition consistency. The paper's key finding is that this multi-task fine-tuning paradigm allows the moderately-sized VLM to establish new state-of-the-art performance, surpassing not only larger, generalist zero-shot models but also highly specialized, purpose-built HMER models on a range of standard benchmarks.

(b) The paper's primary strength, as noted by the reviewers, is its systematic and well-motivated approach to adapting a generalist VLM for a complex, specialized domain. The three proposed fine-tuning tasks were consistently praised as being relevant, creative, and adaptable to other fields. The experimental validation is another major strength; the paper demonstrates state-of-the-art performance against a comprehensive set of baselines and includes a carefully designed ablation study that clearly attributes performance gains to the specific tasks. The work is also highly regarded for its clarity, reproducibility, and the practical utility of its multi-task training setup, which provides a strong new baseline for the HMER community.

(c) The main initial weakness identified by reviewers was the perceived limited novelty, as the core idea involves fine-tuning an existing VLM rather than proposing a new model architecture. Some reviewers also raised concerns that the method's effectiveness appeared heavily dependent on access to in-domain training data, potentially limiting its out-of-the-box generalization. Other initial critiques included the potential computational expense of the Error-Driven Learning approach and a lack of clarity in some of the experimental details and future work descriptions, though most of these points were successfully addressed during the rebuttal period.

(d) Recommendation: Accept (Spotlight)
The paper is a clear Accept due to its technical soundness, state-of-the-art results, and an exemplary rebuttal that addressed all major reviewer concerns. It warrants a Spotlight because it represents a compelling case study for a significant paradigm shift in specialized recognition tasks. While many papers apply large models to new tasks, this work stands out for three reasons. First, it successfully argues that in the era of VLMs, innovation is shifting from bespoke architectures to intelligent, data-centric fine-tuning paradigms, and provides a highly effective and generalizable recipe for doing so. Second, the paper demonstrates strong rigor in ablation studies, particularly through the comprehensive "odd-one-out" generalization experiments and detailed error analysis conducted during the rebuttal, setting a high standard for empirical validation. Finally, it delivers a high-impact practical outcome: a unified model that is not only more accurate but also—counter-intuitively—faster at inference than previous smaller, specialized models, providing a powerful and accessible new baseline that will likely drive future research in the HMER field and Handwriting Recognition field in general.